# Regulatory sites of CaM-sensitive adenylyl cyclase AC8 revealed by cryo-EM and structural proteomics

Basavraj Khanppnavar [1,2,5], Dina Schuster[1,2,3,5], Pia Lavriha [1], Federico Uliana [4], Merve Özel[1], Ved Mehta [1], Alexander Leitner [3], Paola Picotti [3] & Volodymyr M Korkhov [1,2]✉

## Abstract

**Membrane adenylyl cyclase AC8 is regulated by G proteins and calmodulin (CaM), mediating the crosstalk between the cAMP pathway and $Ca^{2+}$ signalling. Despite the importance of AC8 in physiology, the structural basis of its regulation by G proteins and CaM is not well defined. Here, we report the 3.5 Å resolution cryo-EM structure of the bovine AC8 bound to the stimulatory Gαs protein in the presence of $Ca^{2+}$/CaM. The structure reveals the architecture of the ordered AC8 domains bound to Gαs and the small molecule activator forskolin. The extracellular surface of AC8 features a negatively charged pocket, a potential site for unknown interactors. Despite the well-resolved forskolin density, the captured state of AC8 does not favour tight nucleotide binding. The structural proteomics approaches, limited proteolysis and crosslinking mass spectrometry (LiP-MS and XL-MS), allowed us to identify the contact sites between AC8 and its regulators, CaM, Gαs, and Gβγ, as well as to infer the conformational changes induced by these interactions. Our results provide a framework for understanding the role of flexible regions in the mechanism of AC regulation.**

**Keywords** Adenylyl cyclase; Calmodulin; Heterotrimeric G protein; cryo-Electron Microscopy (cryo-EM); Structural Proteomics
**Subject Categories** Membranes & Trafficking; Signal Transduction; Structural Biology

## Introduction

Membrane-integral adenylyl cyclases (ACs) catalyse the conversion of ATP into a universal second messenger, cyclic AMP (cAMP), and play a significant role in transmitting information from sensory cell surface receptors, including G protein-coupled receptors (GPCRs), to multiple effector proteins such as protein kinase A or cyclic nucleotide-gated ion channels (Khannpnavar et al, 2020; Ostrom et al, 2022). Membrane ACs (AC1-9) share a common structural blueprint: each is a polytopic membrane protein with twelve transmembrane helices and two catalytic domains, C1 and C2 (Khannpnavar et al, 2020). They show ~50% homology (Sunahara et al, 1996). The modes of regulation and tissue expression vary for each AC type (Defer et al, 2000). All isoforms are activated by the stimulatory G protein alpha (Gαs) subunit (Iyengar, 1993), and some isoforms (AC1, AC5 & AC6) are inhibited by the inhibitory G protein alpha (Gαi) subunit (Taussig et al, 1993). Gβγ subunits have been shown to modulate the cyclase activity in an AC isoform-specific manner, with some isoforms (AC2, AC4, AC5, AC6, AC7) being activated (Bayewitch et al, 1998; Gao and Gilman, 1991; Gao et al, 2007; Tang and Gilman, 1991) and some isoforms (AC1, AC3, AC8) being inhibited by Gβγ (Diel et al, 2006; Steiner et al, 2006; Tang and Gilman, 1991; Tang et al, 1991), with AC9 not sensitive to Gβγ (Tang and Gilman, 1991).

The molecular basis of the conversion of ATP to cAMP by membrane ACs was revealed by X-ray crystallographic studies of the chimeric catalytic domains of $AC5_{C1}/AC2_{C2}$ (Tesmer and Sprang, 1998; Tesmer et al, 1997; Tesmer et al, 1999). These studies defined the structural basis of how ACs perform a two-metal catalysis reaction, how they are activated by a Gαs subunit and how the plant-derived small molecule forskolin activates the enzyme. Forskolin and Gαs can activate membrane ACs independently or synergistically through stabilising the interaction between the catalytic domains C1a and C2a (Tesmer et al, 1997). The mechanism of regulation by other G protein subunits, such as Gβγ, is not fully understood, partly due to the lack of structural data. Peptide-based inhibition studies have suggested that Gβγ binds the C1/C2 catalytic domains of AC2 (Chen et al, 1995), AC5 (Wittpoth et al, 1999), and AC6 (Brand et al, 2015). In addition, Gβγ has been reported to interact with the N-termini and/or the C1b domain of different AC isoforms, including AC1 (Wittpoth et al, 1999), AC2 (Diel et al, 2006), AC5 (Brand et al, 2015; Wittpoth et al, 1999), and AC6 (Brand et al, 2015; Gao et al, 2007).

The $Ca^{2+}$/CaM-sensitive ACs, AC1 (Westcott et al, 1979), AC3 (Choi et al, 1992), and AC8 (Westcott et al, 1979) play a pivotal role in cellular signalling (Ostrom et al, 2022; Sanchez-Collado et al, 2021). They mediate the crosstalk between distinct signalling pathways (i.e. the GPCR/cAMP pathway and the $Ca^{2+}$ signalling pathway), integrating the different signalling inputs and fine-tuning the cellular responses to them (Sanchez-Collado et al, 2021). The

[1]Laboratory of Biomolecular Research, Division of Biology and Chemistry, Paul Scherrer Institute, Villigen, Switzerland. [2]Department of Biology, Institute of Molecular Biology and Biophysics, ETH Zurich, Zurich, Switzerland. [3]Department of Biology, Institute of Molecular Systems Biology, ETH Zurich, Zurich, Switzerland. [4]Department of Biology, Institute of Biological Chemistry, ETH Zurich, Zurich, Switzerland. [5]These authors contributed equally: Basavraj Khanppnavar, Dina Schuster. ✉E-mail: volodymyr.korkhov@psi.ch

effect of $Ca^{2+}$ on ACs is mediated by the $Ca^{2+}$-sensor protein CaM (Berridge et al, 2000; Raffaello et al, 2016). Binding of CaM to $Ca^{2+}$/CaM-sensitive ACs stimulates their catalytic activity in a $Ca^{2+}$-dependent manner. AC8 was shown to interact with CaM via two distinct conserved calmodulin-binding domains: an amphipathic CaM-binding motif at the N-terminus and an IQ-like CaM-binding motif in the cyclase C2b domain (Fig. 1A) (Herbst et al, 2013; Masada et al, 2009). Based on in vitro studies, the N-terminal CaM-binding motif was proposed to contribute to the constitutive tethering of CaM to AC8, while not being absolutely required for stimulation of the catalytic activity (Gu and Cooper, 1999; Simpson et al, 2006; Smith et al, 2002). The CaM-binding motif on the AC8 C2b domain, on the other hand, was suggested to be critical for activation of AC8 by $Ca^{2+}$/CaM (Herbst et al, 2013; MacDougall et al, 2009; Masada et al, 2009). Similar to AC9, the deletion of the C2b domain results in constitutive superactivation of AC8, and it has been proposed that the $Ca^{2+}$/CaM complex may be involved in the regulation of the autoinhibitory mechanism of AC8 (Herbst et al, 2013; MacDougall et al, 2009; Masada et al, 2009).

We have previously determined the high-resolution structure of the full-length bovine AC9, revealing molecular details of AC9 autoinhibition by its C-terminal domain C2b (Qi et al, 2019). Despite the availability of a homologous AC9 structure and existing biochemical data on AC8, our understanding of the molecular mechanisms of $Ca^{2+}$/CaM-sensitive AC regulation by CaM and G protein subunits is incomplete. To address this knowledge gap, we determined the structure of purified bovine $Ca^{2+}$/CaM-sensitive AC8, complemented by mass spectrometry-based structural proteomics approaches (XL-MS and LiP-MS) and functional studies. The combination of these techniques allowed us to gain deeper insights into the regulation of AC8 and better understand the role of its structured and flexible domains.

## Results

### Purification and biochemical characterisation of bovine AC8

AC8 is known to interact with and to be regulated by CaM, Gαs, Gβγ and forskolin (Fig. 1A) (MacDougall et al, 2009; Masada et al, 2009; Ostrom et al, 2022; Tang and Gilman, 1991; Westcott et al, 1979). To gain insight into the structure, function and regulation of AC8 (Fig. 1A), we expressed bovine AC8 in human embryonic kidney 293 (HEK293) GnTI⁻ cells using a tetracycline-inducible expression system, purified the protein in detergent by affinity chromatography (Appendix Fig. S1), and assessed functional integrity of the preparation with AC activity assays (Fig. 1B,C). Quantitative mass-spectrometry (MS) -based assessment of the purified AC8 preparations determined that endogenously expressed CaM is a negligible contaminant in our AC8 preparations (AC8:CaM ratio of ~500:1; Appendix Fig. S2; Appendix Table S1).

The purified AC8 was efficiently activated by its endogenous activators CaM and Gαs, as well as by the small molecule activator forskolin (Fig. 1D–F). The $EC_{50}$ of CaM-activation was 4.1 nM and 86.6 nM in the presence and absence of $CaCl_2$, respectively (Fig. 1E). Thus, our in vitro experiments using full-length AC8 confirm that CaM-dependent activation of AC8 is a $Ca^{2+}$-dependent process, as has been observed previously (MacDougall et al, 2009; Masada et al, 2009;

Westcott et al, 1979). Fluorescence-detection size-exclusion chromatography (FSEC) binding assays using purified AC8 and a C-terminally yellow fluorescent protein (YFP) tagged CaM (CaM-YFP) showed that the apparent affinity of CaM-YFP for AC8 is ~70 nM in the presence of $Ca^{2+}$ (Fig. EV1). In the presence of a $Ca^{2+}$ chelating agent, EGTA, high affinity binding could not be observed in the FSEC experiment. The difference between the AC8 activation $EC_{50}$ values and the apparent $K_d$ values in the binding assays likely stems from the use of a fluorescently tagged CaM and the non-equilibrium FSEC separation technique. Nevertheless, the adenylyl cyclase activity assays combined with the binding assays clearly indicate that in the presence of $Ca^{2+}$ CaM binds AC8 with nanomolar affinity.

The GTPγS-bound Gαs activated AC8 with high affinity ($EC_{50}$ of 25.5 nM; Fig. 1F). AC8 was inhibited by $Gβ_1γ_2$ subunits, consistent with previous observations (Tang and Gilman, 1991). Interestingly, the $EC_{50}$ for the activation of AC8 by Gαs is 30-fold lower than for bovine AC9, suggesting an isoform-dependent affinity. In contrast to CaM and Gαs, Gβγ inhibits AC8 with lower micromolar affinity with an $IC_{50}$ of 0.4 µM (Fig. 1E,F). Forskolin activates AC8 with nanomolar affinity ($EC_{50}$ = 136 nM) (Fig. 1D). The results are in line with the available body of evidence (MacDougall et al, 2009; Masada et al, 2009; Ostrom et al, 2022; Tang and Gilman, 1991; Westcott et al, 1979), confirming that our purified AC8 preparations represent a native-like state of the protein, with intact sensitivity to the known regulators.

### Cryo-EM structure of the AC8-CaM-Gαs complex

To structurally characterise AC8, we used single-particle cryo-EM analysis of the protein in the presence and in the absence of $Ca^{2+}$/CaM, Gβγ, and Gαs. The concentrations of each of the protein components (AC8, CaM, Gαs) in the cryo-EM samples were controlled by mixing the purified proteins at specific molar ratios immediately prior to grid freezing. Among the tested combinations, one sample proved to be suitable for structure determination: AC8 in the presence of $Ca^{2+}$/CaM, activated Gαs, MANT-GTP, and forskolin. After multiple rounds of 2D and 3D classifications, the best set of particles resulted in a 3D reconstruction at a resolution of 3.5 Å (Appendix Figs. S3–S5, Appendix Table S2).

The structure of AC8-CaM-Gαs (AC8 bound to Gαs in the presence of $Ca^{2+}$/CaM) revealed most of the features of AC8, including the transmembrane (TM) domain, the helical domain (HD), the catalytic domain (CAT), as well as the G protein Gαs subunit (Fig. 1G–I). The N-terminus (1–164), the C-terminus (1165–1251) and the C1b domain (585–670) could not be resolved in the final 3D reconstruction.

### The AC8-Gαs interface

The stimulatory G protein subunit Gαs activates each of the membrane AC isoforms. We compared the features of the AC8-Gαs interaction revealed by our structure with previously reported structures: the cryo-EM structure of AC9-Gαs (Qi et al, 2022; Qi et al, 2019) and the X-ray crystallography structure of the $AC2_{(C2a)}$-$AC5_{(C1a)}$-Gαs complex (Tesmer et al, 1999). Each of these ACs interacts with Gαs via conserved and non-conserved polar residues (Fig. EV2, Appendix Table S3). We could also observe differences in the number and type of interfacial residues (Fig. EV2) which are not necessarily involved in direct interactions with Gαs (except for a few

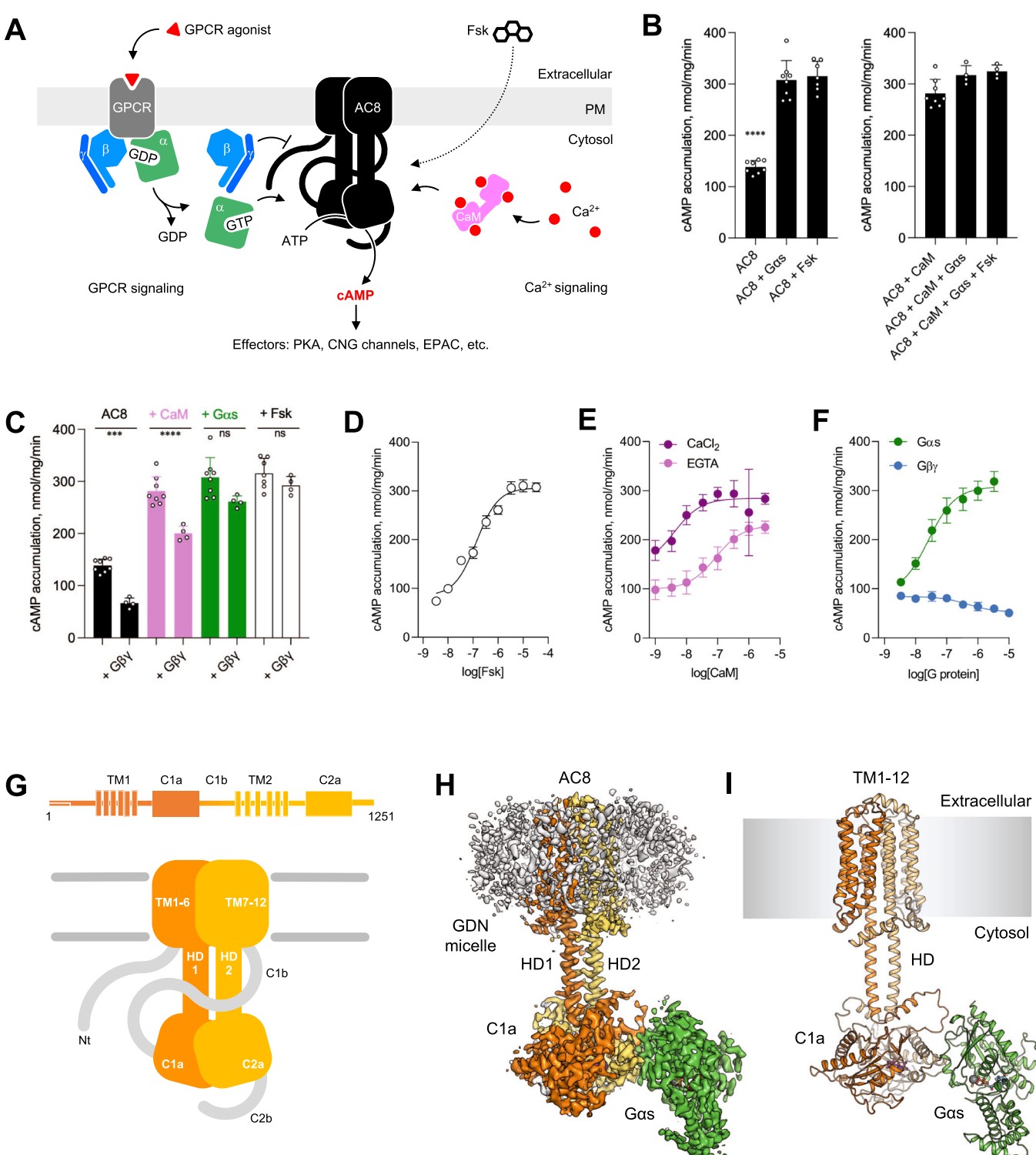

hydrophobic contacts) but may influence the AC-G protein binding. Interestingly, the apparent affinity of Gαs for AC8 (EC$_{50}$ of 25.5 nM, Fig. 1F) is higher than that for AC9 (EC$_{50}$ of 802.9 nM) (Qi et al, 2022). The difference in affinity between Gαs and different AC isoforms is likely explained by the divergent interfacial interactions, which may play a role in a complex cellular environment where specific signalling inputs may be required, e.g., in the presence of several co-expressed AC isoforms.

## Conformational heterogeneity in the substrate binding site of AC8

The structure showed a clear density corresponding to forskolin (Fig. 2A–C, Appendix Fig. S4). In stark contrast, the density for MANT-GTP was not well resolved (Fig. 2A–C, Appendix Fig. S4). We tested whether this could be explained by a drop in affinity for MANT-GTP in the presence of forskolin, but the affinity of AC8

**Figure 1.  Biochemical characterisation of purified bovine AC8.**

(**A**) Schematic diagram depicting the key players in regulation of AC8 and its role in mediating the crosstalk between the cAMP and $Ca^{2+}$ signalling pathways; Fsk forskolin, CaM calmodulin, PM plasma membrane. (**B, C**) Enzymatic activity of AC8 in the presence of various interacting partners (CaM, Gαs, Gβγ and forskolin. The AC8 control bars ("AC8", "AC8 + CaM", "AC8 + Gαs", "AC8 + Fsk") are identical in b and c. For all experiments, the data are shown as mean ± standard error of the mean (SEM) ($n = 4$; for AC8 control bars, $n = 8$; technical replicates). One data point was excluded for AC8 vs FSK (replicate 4 in panel (**B**), and associated replicate 8 in panel (**C**)) as an identified outlier (ROUT method; Q = 0.1%). Statistical significance was assessed using one-way ANOVA, followed by Tukey's multiple comparisons test, with $P < 0.0001$ ("*****"), $P = 0.0006$ ("***", **C**), $P = 0.0385$ ("*", **C**). No significant difference is indicated as "ns" (**C**). (**D–F**) Dose-response curves for AC8 activity in the presence of various interaction partners. For all experiments, the data are shown as mean ± S.D. ($n = 3$, for AC8-Gαs, $n = 4$). (**G**) Pictographic representation of membrane adenylyl cyclase AC8 topology. (**H, I**) Cryo-EM map (**H**) and model (**I**) of the AC8-Gαs-$Ca^{2+}$/CaM-Forskolin-MANT-GTP complex. Source data are available online for this figure.

for MANT-GTP was not affected by the presence of either forskolin or other activators (Fig. 2D,E). We previously observed that AC9 is capable of binding MANT-GTP in different conformations dependent on the presence of forskolin (Qi et al, 2022). These observations are in contrast to the distinct single conformation of MANT-GTP observed in the crystal structure of the chimeric catalytic domain of AC2$_{(C2a)}$-AC5$_{(C1a)}$ (Tesmer et al, 1999). Thus, the behaviour of the nucleotide-binding site in AC8 is distinct from that of AC9 or the chimeric AC2$_{(C2a)}$-AC5$_{(C1a)}$.

To ensure that the observed conformation is not induced by the detergent micelle, we reconstituted AC8 in MSP1E3D1 nanodiscs with native brain polar lipids and prepared a cryo-EM sample using similar conditions (Appendix Fig. S6A,B). As the density of MANT-GTP was not well resolved in the detergent-purified structure, we replaced MANT-GTP with ATPαS. We reasoned that ATPαS should be a better substrate surrogate than the nucleotide-based inhibitor MANT-GTP since it is structurally similar to the substrate ATP. The lipid nanodisc sample was processed similarly to the detergent sample. After multiple rounds of 2D and 3D classifications, the 3D reconstruction resulted in a 3.97 Å map of the AC8-Gαs complex and a 3.5 Å resolution structure after focused refinement of the AC8 catalytic domain-Gαs complex. The conformation of the nanodisc reconstituted AC8 is similar to the conformation observed in detergent. The N-terminus, C-terminus, C1b, and CaM remain unresolved (Appendix Figs. S6–8). The density for ATPαS was also not well resolved in this structure (Fig. 2F; Appendix Fig. S8). Together with the lack of a well-resolved density of MANT-GTP, these data are indicative of substrate and inhibitor conformational heterogeneity.

## Specificity of forskolin binding to AC8 and other membrane ACs

The forskolin-binding interfaces of AC8, AC9, and AC2$_{(C2a)}$-AC5$_{(C1a)}$ show both conserved and non-conserved interfacial residues (Fig. 2G–K). The interfacial residues closer to the ATP-binding site are highly conserved, whereas the residues near the Gαs binding site exhibit some variability (Fig. 2G). These minor differences may contribute to the observed differences in the affinity of forskolin for the various AC isoforms (Erdorf et al, 2011; Seth et al, 2022), and may also underlie the Gαs-dependent conditional activation of AC9 (Baldwin et al, 2019; Qi et al, 2019). Comparisons of the forskolin binding sites of AC8, AC9, and AC2$_{(C2a)}$-AC5$_{(C1a)}$ show that functional differences between these ACs are likely to be explained by four residues (in AC8: W534, D537, A994, and L1010) (Fig. 2H–K). All four

are in close proximity of forskolin in one or more of the available AC structures. One residue in particular, L1010, stands out: it is conserved among all ACs but one, AC9. In AC9 this position is occupied by Y1083, which directly faces forskolin with its hydroxyl group (Fig. 2H–K). AC9 alone is not stimulated by forskolin and can only be activated by forskolin in the presence of a bound Gαs subunit (Baldwin et al, 2019; Qi et al, 2022). It is thus likely that a leucine residue in this key position (L1010 in AC8) favours G protein-independent forskolin activation of a membrane AC, whereas the presence of a tyrosine residue (Y1083 in AC9) may require G protein to enable forskolin-mediated AC activation.

## The TM domain of AC8 features a putative pocket for extracellular ligands

Comparison of the structure of AC8 with the previously solved full-length structures of the membrane ACs, AC9 and the mycobacterial membrane AC Rv1625c/Cya, showed differences in the degree of rotation between the CAT and TM domains (Fig. 3A). Nevertheless, the AC8 TM domain adopts a similar 12 TM helix arrangement to that of AC9, and a similar organisation of TM1-5/TM7-11 to TM1-5 of Cya (Fig. 3B–D). The observed differences in the relative orientation of the TM domain and CAT with respect to each other, after 3D alignment of AC8 and its homologues, seems to originate from the coiled-coiled HD helices of the ACs (Fig. 3A). The HDs of AC8 and AC9 are substantially longer than HD of Cya, and the difference in orientation of the TM domains relative to the CAT domains in AC8 and AC9 may be due to small structural changes in the HDs.

The extracellular region (except for residues 811-831) of the AC8 TM domain is better resolved compared to that in AC9 (Fig. 3E,F; Appendix Fig. S4). This near-complete structure of the extracellular side of the TM domain reveals a prominent cavity formed at the interface of the TM1-7 helices (Fig. 3G). The pocket is formed by the shorter TM4 helix and loop between TM3-4 (ECL2, Fig. 3E–H, Appendix Fig. S4) and is partially negatively charged, reminiscent of Cya (Fig. 3I) (Mehta et al, 2022). Our previous study of Rv1625c/Cya (Mehta et al, 2022) indicated that the membrane anchor of this mycobacterial membrane AC may have a receptor function for cations, such as divalent metals. The features of the extracellular portion of the TM region in AC8 are compatible with a similar function, i.e., interactions with positively charged ligands (ions, small molecules, peptides or proteins). Further studies will be required to determine whether the extracellular surface of AC8 may contain functionally relevant ligand-binding sites.

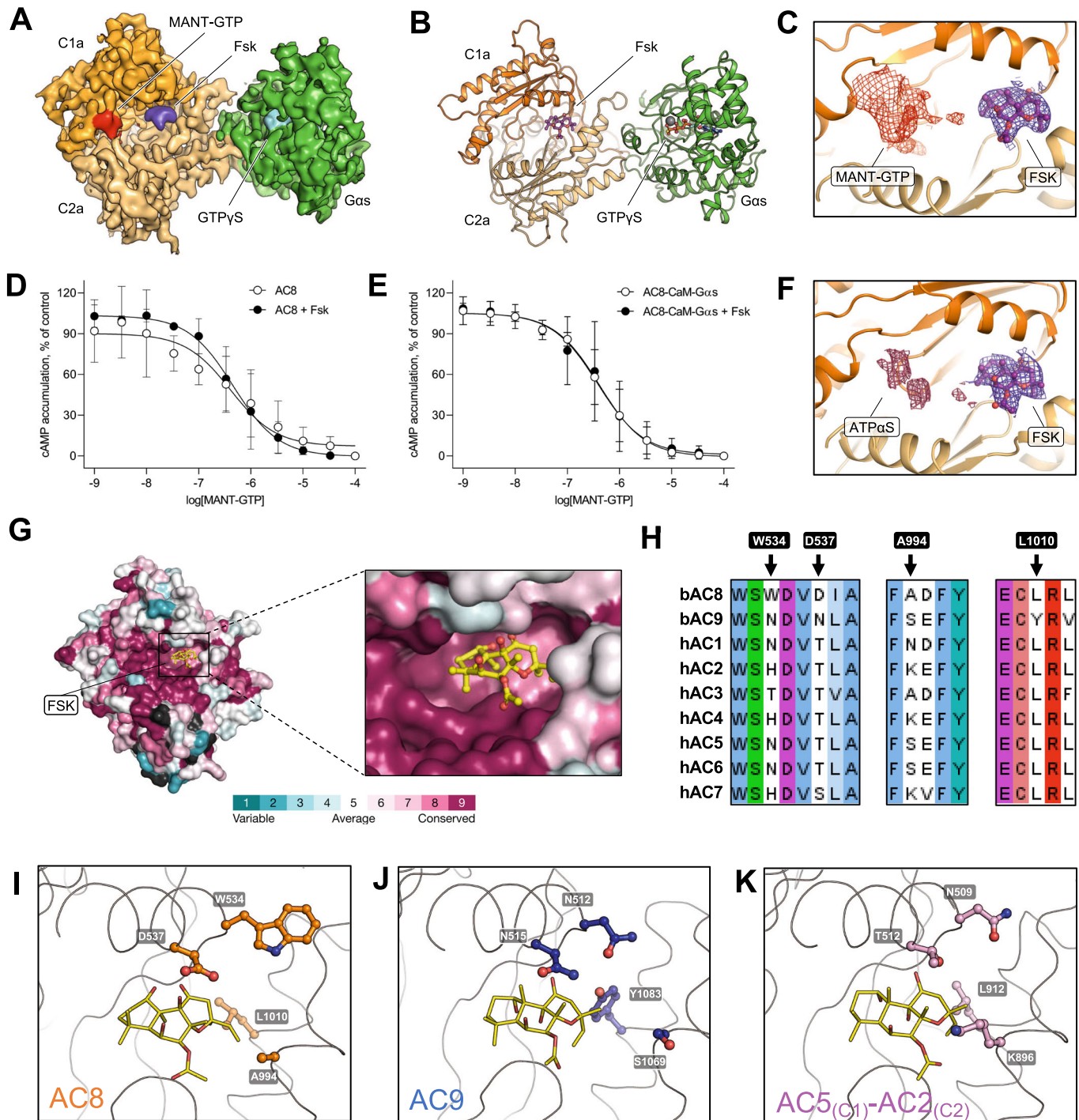

**Figure 2.   ATP- and forskolin-binding pockets in AC8.**

(A,B) Cryo-EM map and model of the catalytic domain of AC8 in complex with Gαs, $Ca^{2+}$/CaM, forskolin and MANT-GTP. (C) Electron density in the substrate-binding pocket of AC8 shows a well-defined density for forskolin and a poorly resolved density for MANT-GTP. (D, E) Dose-response curves of MANT-GTP in the presence and absence of forskolin, CaM, and Gαs. For all experiments, the data are shown as mean ± S.D. ($n = 3$; technical replicates). One-way ANOVA followed by Tukey's multiple comparisons tests showed that all logIC$_{50}$ values are not significantly different from each other. (F) Unresolved density in the ATP-binding site of the AC8-Gαs-$Ca^{2+}$/CaM-forskolin-ATPαS complex solved in lipid nanodiscs. (G) Conservation scores for nine membrane AC isoforms computed with ConSurf (Ashkenazy et al, 2016). (H) A segment of multiple sequence alignment showing key non-conserved residues in forskolin binding interfaces in nine membrane AC isoforms. (I–K) Comparison of forskolin-binding interfaces in AC8, AC9 (PDB ID: 6R40), and the chimeric AC2$_{(C2a)}$-AC5$_{(C1a)}$ complex (PDB ID: 1TL7). The interfacial residues in panels (H–K) were chosen based on their proximity (within 4 Å) to forskolin in any one of the AC isoforms. Source data are available online for this figure.

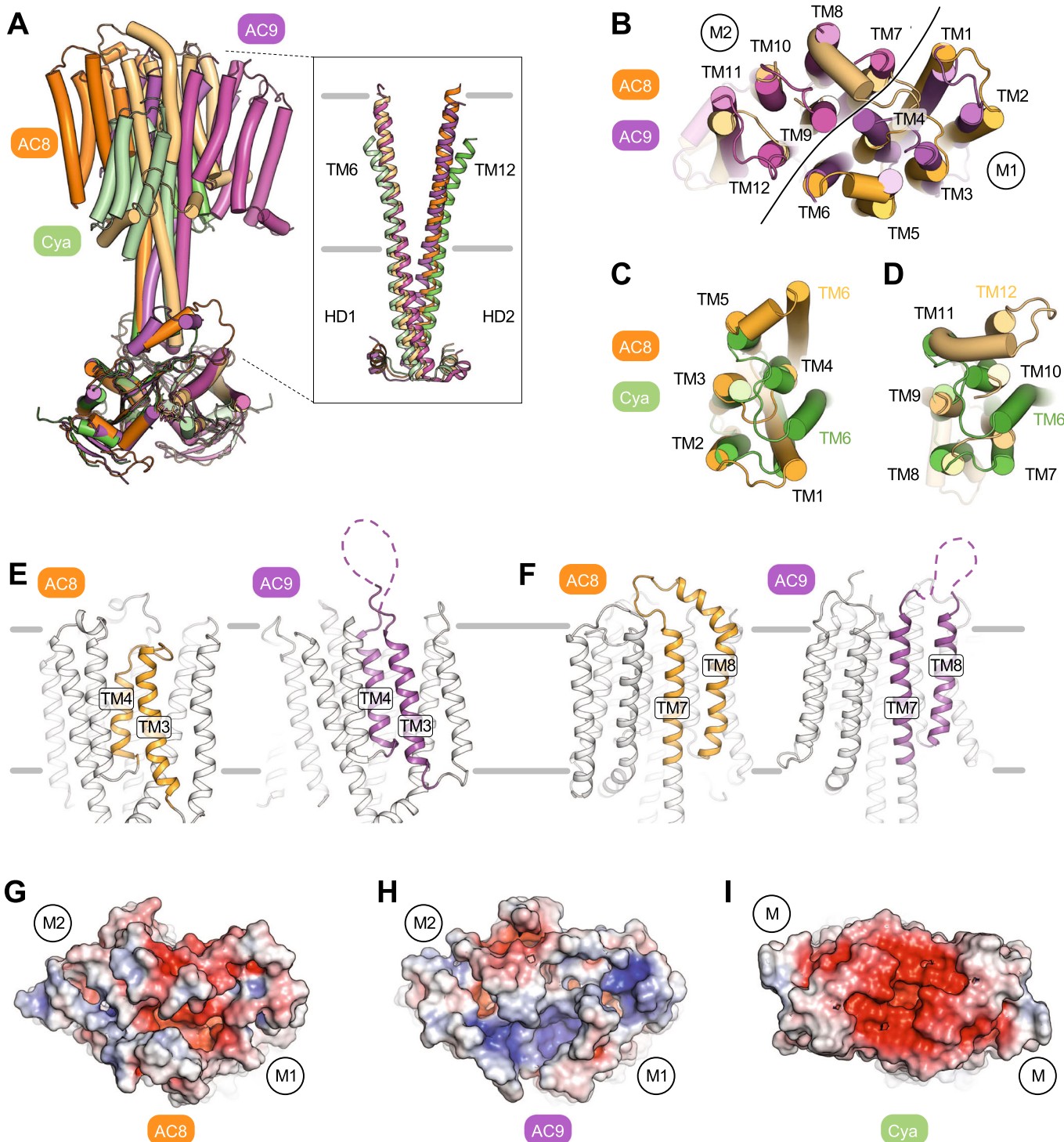

**Figure 3. Key structural features of the transmembrane domain of AC8.**

(A) Comparison of cryo-EM structures of AC8, AC9 and *M. tuberculosis* Rv1625c/Cya. The membrane domains of these ACs are tilted differently relative to the helical and catalytic domains. The insert shows the alignment of the helical domains of the three membrane ACs of the known structure. (B–D) Alignment of the membrane domains of AC8 and AC9 (B) and AC8 and Cya (C, D). The line in (B) indicates the boundaries of the two hexahelical domains ("M1" and "M2"). (E, F) Depiction of the key structural differences at the extracellular side of the AC8 TM domain compared to AC9, in TM3-4 (short loop in AC8, disordered loop in AC9) and in TM7-8 (ordered folded loop in AC8, disordered extended loop in AC9). (G–I) An extracellular view of the TM domains of AC8, AC9, and Rv1625c/Cya showing the potential extracellular ligand binding pockets in surface representation, coloured according to electrostatic potential. The "M1" and "M2" labels indicate the positions of the 6-TM bundles of AC8 (G) and AC9 (H), "M" denotes the 6-TM bundles of Rv1625c/Cya (I).

## LiP-MS reveals involvement of flexible domains in activating and inhibiting interactions of AC8

Limited proteolysis-coupled mass spectrometry (LiP-MS) has previously been demonstrated to detect protein-binding interfaces upon addition of a protein interactor of interest to a complex proteome or membrane extract, due to changes in protease accessibility upon protein-protein interaction (Holfeld et al, 2023). We therefore used LiP-MS to detect binding interfaces and other regions of altered surface accessibility upon interaction of AC8 with Gαs and Gβγ. We also reanalysed a previously published LiP-MS dataset of the interaction of AC8 with CaM, conducted under the same experimental conditions (PRIDE ID: PXD039520; (Holfeld et al, 2023)). Since binding interfaces of AC8-CaM were previously known, and since we determined those of AC8-Gαs experimentally by cryo-EM, we could use these orthogonal datasets for cross-validation.

We prepared crude membrane suspensions of AC8-YFP-overexpressing HEK 293 GnTI⁻ cells and titrated them with interactors of interest in increasing concentrations. We performed limited proteolysis with proteinase K on all samples under native conditions, followed by denaturation and complete trypsin digestion. We controlled for protein abundance changes by analysing control samples subjected only to the trypsinisation step under denaturing conditions after treatment with the highest or lowest interactor concentrations (see Methods). We measured all samples with data-independent acquisition and fit four-parameter dose-response curves onto the measured peptide profiles. To determine regions changing in protease accessibility, we filtered for peptides with a Pearson correlation r of >0.85 between the measured peptide intensities and the fitted dose-response model (Appendix Fig. S9). Peptides with a correlation r < 0.85 were considered unchanged (Fig. 4D). This threshold (0.85) was defined based on the validation AC8-CaM and AC8-Gαs data sets, for which the binding interfaces were known. We detected 167 AC8-YFP peptides in the CaM dataset, 177 in the Gβγ dataset, and 180 in the Gαs dataset, corresponding to a sequence coverage of AC8 of 55-60%. As expected, transmembrane domains of AC8 were not detected in any of the datasets. Nine AC8 peptides showed evidence of a change in protease accessibility upon addition of CaM (Fig. 4A), six peptides upon addition of Gαs (Fig. 4B), and six different peptides upon addition of Gβγ (Fig. 4C).

The changes induced by the different interactors were located on different AC8 domains. CaM-induced accessibility changes on the AC8 N-terminus and C2b domain (C-terminus), at the known binding interfaces (Gu and Cooper, 1999; MacDougall et al, 2009) (Fig. 4A). Interestingly, the significantly changing peptides show profiles with opposite directionality. We find that peptides closer to the CaM binding motifs increase in abundance, while peptides further away decrease in abundance. Assuming that the peptides at the binding sites increase in abundance due to increased protection from proteolytic cleavage, the regions further away likely become more accessible to proteolytic cleavage and therefore we detect a decrease in their abundance upon CaM binding. Addition of Gαs to AC8 induced changes in both catalytic domains C1a and C2a, with the one peptide at the C-terminal end of C2a, overlapping with the first amino acids of C2b. The detected accessibility changes cover the binding interface we could resolve in our cryo-EM structure on C2a (residues 1073–1081), as well as an extension of this region

(1083–1106), and a neighbouring peptide (1132–1142) (Fig. 4B). We also identify an accessibility change on C1a (470–477) that could result from the Gαs N-terminus that is not resolved in our cryo-EM structure. Addition of Gβγ changed the surface accessibility of all three flexible AC8 domains (N-terminus, C1b and C2b domains) and the YFP-tag (Fig. 4C). The changes detected upon interaction with Gβγ are distinct from the changes induced by CaM or Gαs. Our data shows that CaM, Gαs and Gβγ interact with and/or induce conformational changes in distinct domains of AC8, and that these interactions involve flexible regions of the cyclase. Moreover, these results indicate that the AC8 activation mechanisms of CaM and Gαs are distinct from each other and may occur concurrently, in line with the activity data shown in Fig. 1 and consistent with the role of AC8 as an integrator of the G protein and the Ca²⁺ signalling inputs.

## XL-MS provides evidence of AC8 oligomerisation

We used XL-MS to better interpret the changes detected with LiP-MS, in particular, to distinguish conformational changes from binding interfaces and additionally to obtain proximity information. The results of the two crosslinkers we used (dissuccinimidyl suberate (DSS) (Leitner et al, 2014b) and a combination of pimelic acid (PDH) and the coupling reagent 4-(4,6-dimethoxy-1,3,5-triazin-2-yl) (DMTMM) (Leitner et al, 2014a)) are combined and discussed together.

We first examined AC8-AC8 crosslinks (between two separate AC8 molecules), since previous studies have suggested that ACs are able to form homodimers with interaction interfaces located in their catalytic and transmembrane domains (Gu et al, 2002; Tang et al, 1995). Consistent with these results, our XL-MS experiments of AC8 identified several homomultimeric links, indicating self-association of AC8 (Fig. 5B,C). The general architecture and homomultimeric link sites are reproducible across the different screened conditions. These sites are found on structured (C1a, C2a) domains, as well as on the flexible domains (N-terminus, C1b and C2b domains). Docking of a dimeric structure based on the identified self-association links is not possible, indicating either a high flexibility of AC8 dimers or the existence of different dimeric or oligomeric structures. Consistent with these data, the purified AC8 elutes as two peaks (monomer and dimer) in our SEC chromatogram (Appendix Fig. S1). Future work on stabilising or isolating a conformationally homogeneous dimeric AC8 will be necessary to determine the structure of the dimer by cryo-EM. The XL-MS results presented here serve as an important piece of evidence for the existence of dimeric or multimeric AC assemblies in the absence of direct structural evidence.

## XL-MS identifies the direct interaction sites of CaM, Gαs and Gβγ

Our data on purified AC8 (Fig. 5A) show that the unresolved, flexible domains (N-terminus, C1b, C2b) of AC8 can be detected in close proximity to the structured domains. While many of the crosslinks can be explained with the cryo-EM structure of AC8 (Fig. 5A, blue crosslinks), some detected crosslinks between the C1b domain or the helical domain and the catalytic domains cannot be explained with our structure and could indicate an altered position of C1b or high domain flexibility (Fig. 5D,E, red

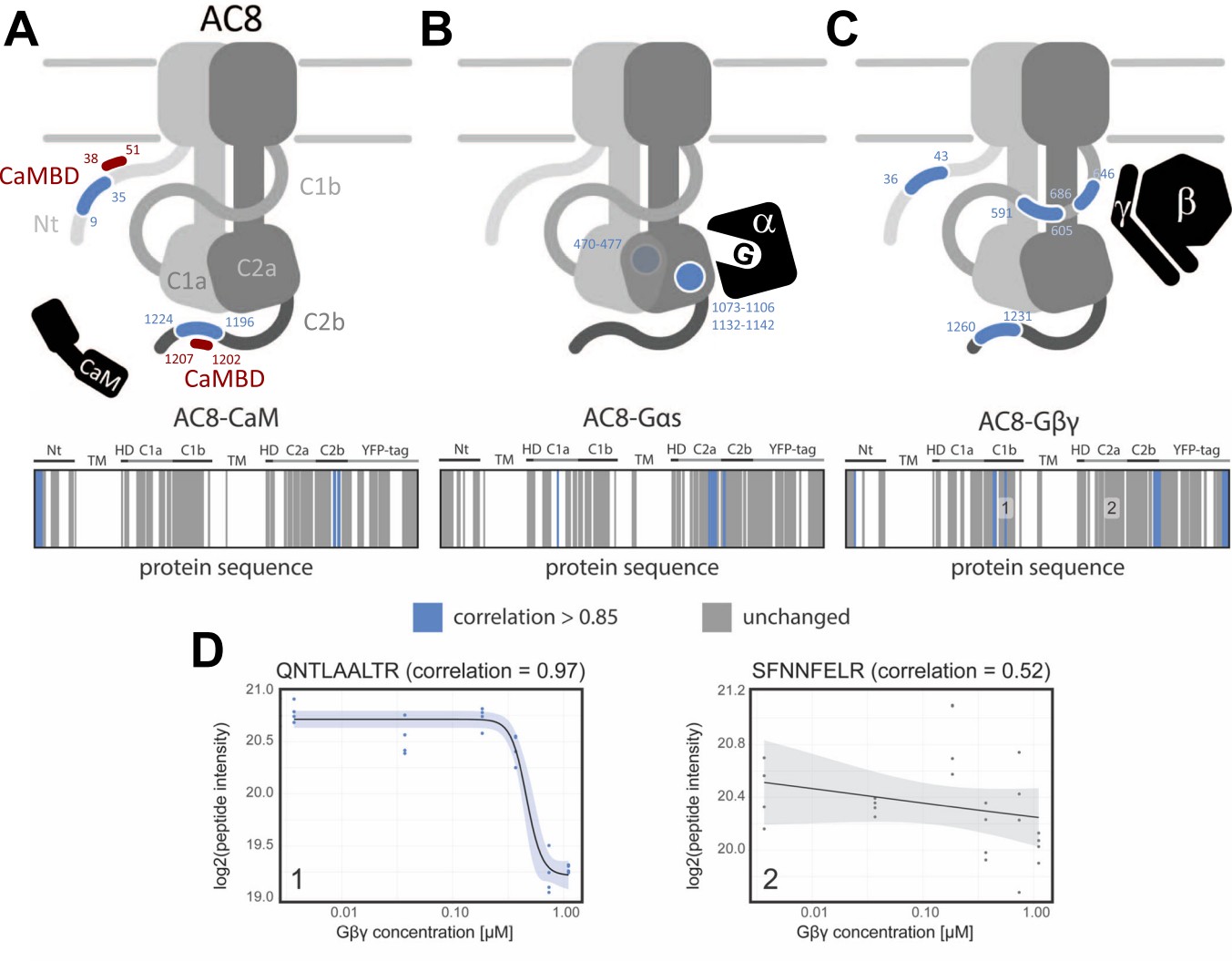

**Figure 4. LiP-MS reveals binding interfaces of AC8 interactors.**

(A–C) Sketches of AC8 structures with detected changes highlighted with blue lines (on flexible regions) or circles (on structured regions). CaM-binding domains are indicated in red. Barcode plots in the middle of the figure show detected peptides in grey and peptides that changed upon addition of an interactor (Pearson correlation r > 0.85) in blue, along the sequence of the protein. (D) Two exemplary interaction curves for peptides detected in the AC8-Gβγ experiment are shown below. The significantly changing peptide is highlighted in blue, the unchanged peptide is shown in grey. Addition of CaM has an effect on protease accessibility of the AC8 N-terminus and the AC8 C-terminus. Addition of Gαs changes the accessibility of regions in the catalytic domains C1a and C2a, with one peptide overlapping between the C-terminal end of C2a and C2b. Gβγ addition changes the accessibility of regions in the three flexible protein domains: the N-terminus, C1b and C2b with additional changes on the YFP-tag.

crosslinks). The violated crosslinks (>30 Å, violation cut-off based on the maximum Cα-Cα distance for DSS (Merkley et al, 2014)) are highly reproducible and occur in all screened conditions (Appendix Fig. S10). Additionally, we observed that Gαs and CaM were copurified with AC8 and were present and interacting with AC8 in all analysed conditions (Figs. 5A, 6).

The addition of CaM results in detection of crosslinks that were not detected when crosslinking AC8 alone. We find changes in the C1a and C2a domains but the links between C1a and C2a are not changed (Fig. EV3A), suggesting that activation by CaM may cause conformational changes within the domains, with little or no change to the catalytic domain assembly.

XL-MS of AC8 in the presence of added Gαs (Fig. 6B,F) shows that the Gαs N-terminus is flexible and has multiple contact sites with the AC8 N-terminus, the C1a domain and the helical domain. The differences between the AC8-Gαs and the AC8-CaM-Gαs complex are only minute: analysis of the crosslinking data for these two samples shows the flexibility of the Gαs N-terminus. In the AC8-CaM-Gαs sample, we observe a crosslink between the Gαs N-terminus and CaM, suggesting that both can be in close proximity to each other. In addition, we detect the same violated crosslink between AC8 and Gαs (Gαs E344xAC8 K581) in the AC8, the AC8-CaM, the AC8-CaM-Gαs and the AC8-Gβγ sample structure (Figs. 6A–D and EV3). This indicates that Gαs may have

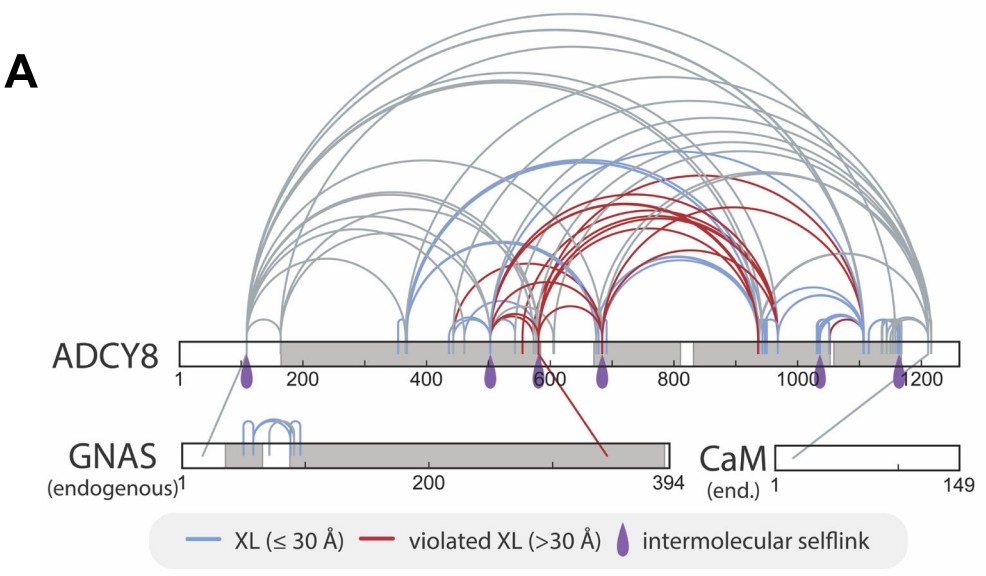

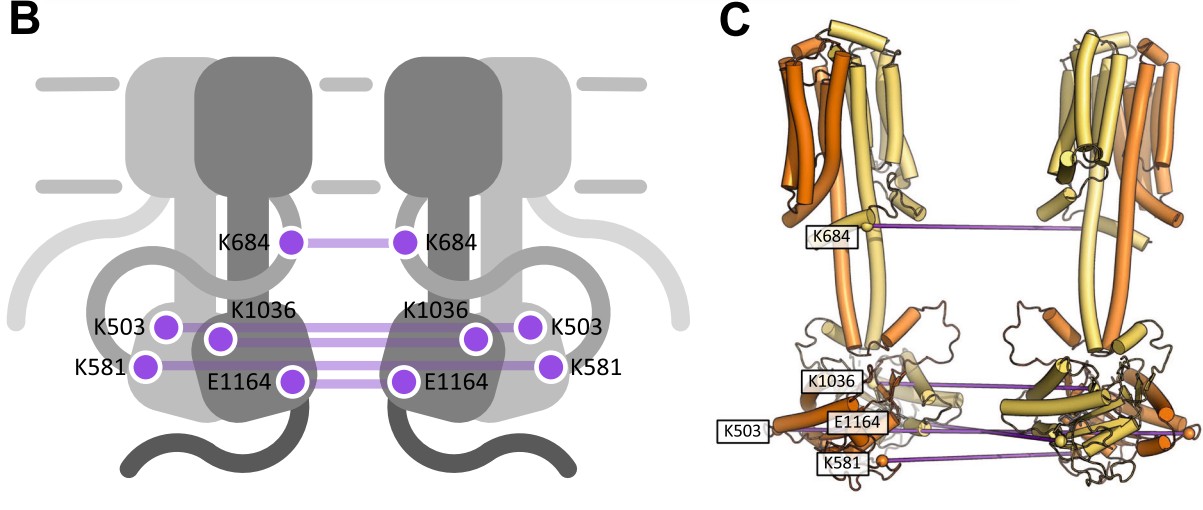

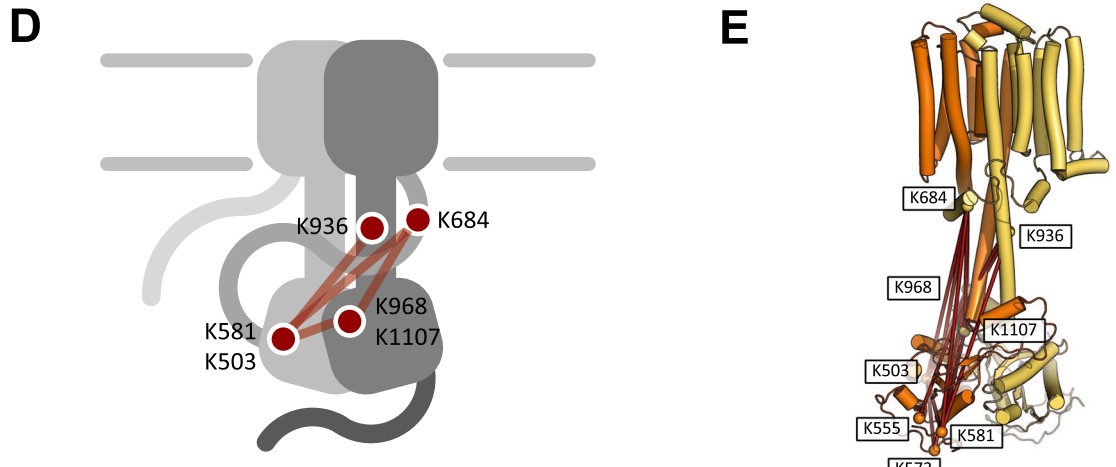

◄ **Figure 5. Intermolecular self-links and crosslinks between AC8 and copurified interactors indicate flexibility in AC8 assemblies.**

(A) XL-MS of AC8 shows self-association sites (homomultimeric links) (purple drops) evident of protein dimerisation. The links are on different sites of the protein that do not fit to only one dimeric structure but could be explained by multiple different dimeric structures. Regions covered in our AC8 structure are highlighted in grey. AC8 is copurified with CaM and Gαs (GNAS). Links with >30 Å distance are considered violated crosslinks (red). Links with <30 Å distance are highlighted in blue. Homomultimeric link sites are indicated as purple drops. The crosslinking map was produced with XiNET (Combe et al, 2015). (B, C) The homomultimeric crosslink positions in the AC8 structure are shown as purple spheres and lines, indicated on sketches. Oligomerisation sites are found on different AC8 domains, indicating that different AC8 complexes can exist. The juxtaposed models in (C) illustrate the crosslink positions. (D, E) Violated crosslinks are shown in the sketch (D) and indicated in the AC8 model (E). The violated crosslinks indicate that there is a lot of structural heterogeneity and flexibility in the sample. Violated crosslinks can also be a result of AC8 oligomerisation. For simplicity of the sketch, only selected violated crosslinks are highlighted, based on their domain location and consistent with the observed trend.

several types of interactions in addition to the one resolved in the cryo-EM structure.

In all analysed samples we could detect crosslinks of CaM with the C2b domain of AC8 (CaM E15xAC8 K1211). The AC8-CaM experiment also shows that CaM can be in close proximity to the AC8 C1a domain (CaM K95xAC8 K503) (Fig. 6A,E), which is not the case in the presence of Gαs (Fig. 6C,G). The AC8-Gβγ experiment showed contact sites of Gβγ with the AC8 N-terminus (Gγ K32 × AC8 K109), the C1b domain (Gβ D312 × AC8 D674, Gβ D312 × AC8 D678), the helical domain (Gβ D312 × AC8 K936) and the C2b domain (Gβ D312 × AC8 E1216) (Fig. 6H). The crosslinks between the resolved parts of C1b, the helical domain and Gβγ facilitated protein docking and showed that Gβγ is positioned close to the helical domain, where it interacts with all three flexible domains (N-terminus, C1b, C2b) (Fig. 6H), further suggested by our LiP-MS experiment (Fig. 4C). When comparing the crosslinks detected in the AC8-Gβγ sample with the AC8 sample, we find additional crosslinks between the helical domain and the C1b domain (Fig. EV4D). This finding supports the hypothesis that the interaction of AC8 with Gβγ causes a rearrangement between the helical domain and the C1b domain. Based on our data, we suggest that the interaction likely involves the AC8 N-terminus, C1b, the helical domain and the C2b domain. Interestingly, a recent cryo-EM study of the AC5-Gβγ complex (Yen et al, 2023) revealed an arrangement of the Gβγ subunits in the complex remarkably close to that observed in our docking model for AC8-Gβγ (Fig. 6I,J, Appendix Fig. S11). As our attempts to perform cryo-EM analysis of AC8-Gβγ indicated that the complex may not be sufficiently ordered for direct structure determination, hybrid approaches are necessary to further characterise the modes of AC binding and regulation by Gβγ subunits.

## Discussion

Our combined cryo-EM and structural proteomics approach applied to AC8 resulted in the following key findings (Fig. 7): (i) the cryo-EM structure of AC8 features a negatively charged extracellular pocket, which may provide an interface for a yet unknown interaction partner; (ii) the forskolin-bound state of AC8 may not be conducive to tight nucleotide binding; (iii) AC8 is likely capable of forming oligomers, in a process that involves the flexible / unstructured protein domains, such as the N-terminus, the C1b domain and the C-terminus; (iv) these flexible domains play a key role in the interactions between AC8 and its regulators (G proteins and CaM).

Contrary to previous studies on AC9, where obtaining a forskolin-bound state of the protein presented a substantial challenge (Qi et al, 2019), we observe a well-resolved density of forskolin bound to AC8, while the densities corresponding to the nucleotides (MANT-GTP or ATPαS), are less well defined. This is surprising, considering the ability of forskolin alone to fully stimulate AC8's cAMP production. The logical assumption we made prior to performing the experiments described here was that a potent activator, such as forskolin, should increase the affinity for the nucleotide. This should coincide with an improvement in the quality of the density corresponding to the bound nucleotide analogue. In contrast to this assumption, the 3D class that resulted in the best 3D reconstruction of AC8 features a well-resolved density for the bound forskolin and a poorly defined density corresponding to the active site (Fig. 7A). One possible interpretation of this finding is that forskolin-binding favours a conformation of AC8 that is compatible with high turnover rate (i.e., binding of ATP, fast conversion of ATP to cAMP, followed by rapid release of the products), but not with high affinity binding of the nucleotide analogues to the active site. It is conceivable that neither MANT-GTP nor ATPαS accurately recapitulate the interaction of ATP with the active site of AC8. We have previously observed the substantial flexibility of the AC9 ATP-binding site that can accommodate MANT-GTP in different poses dependent on the presence of forskolin (Qi et al, 2022). It is thus possible that AC8 binding of forskolin produces a conformation that does not favour a well-defined and uniform nucleotide pose. Furthermore, if a nucleotide-bound conformation of AC8 is not stable or is heterogeneous, the MANT-GTP- or ATPαS-bound states may be "lost" among the particles that are excluded during the 2D or 3D classification steps, as they do not contribute to a well-ordered protein conformation.

Cryo-EM analysis has become one of the primary tools for structure-function studies of proteins and protein complexes. The major caveat of this, and essentially every other structural biology approach, is that it relies on the conformational homogeneity of the sample. An ordered part of the protein can in many (if not most) cases be resolved by single particle analysis. This is the case for the relatively well-ordered portions of AC8: the TM region, the helical domain, the catalytic domains, and the bound G protein subunit. Contrary to this, the unstructured regions of the AC8 present a major challenge for structural studies, as they cannot be resolved by conventional structural biology approaches, evident from the density map of AC8 that lacks the features corresponding to the N-terminus, the C1b and C2b domains. This challenge is not intrinsic to ACs alone since a wide range of proteins are known to be regulated by unstructured domains. However, specifically in the AC field interesting parallels can be drawn in the regulatory mechanisms of AC8 and the bacterial adenylyl cyclase toxins, *Bordetella pertussis* CyaA (O'Brien et al, 2017) and the *Bacillus anthracis* oedema factor (Drum et al, 2002). CaM binding has been found to allosterically influence the dynamics of the catalytic sites

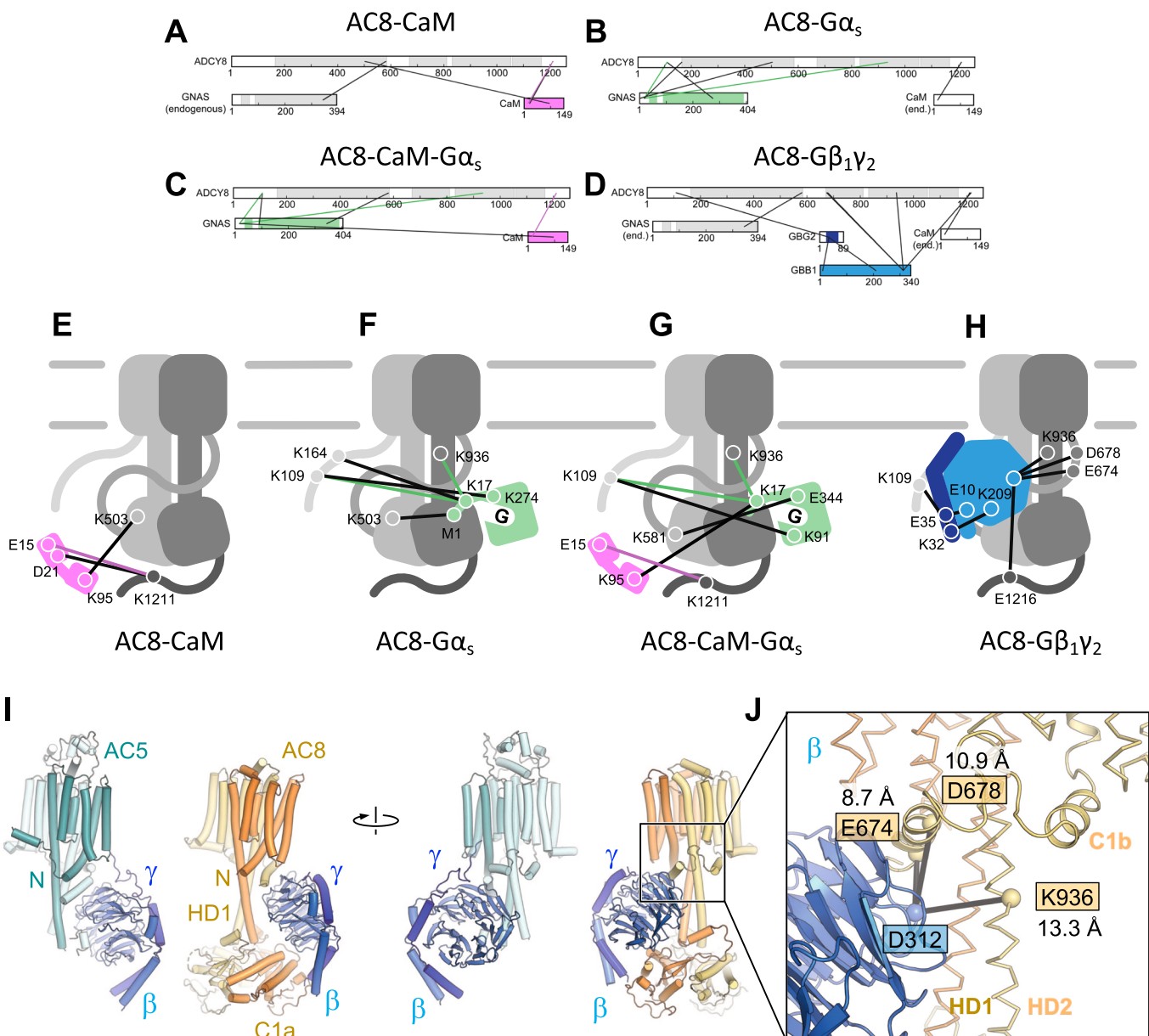

**Figure 6. Crosslinks between AC8 and its interaction partners CaM, Gαs and Gβγ.**

(**A**) CaM interacts with the C2b domain of AC8 and is in close proximity to the C1a domain. Added interactors are coloured (CaM = pink, Gαs = green, Gβγ = blue), copurified interactors are labelled as "endogenous" or "end.". Heteromeric crosslinks (crosslinks between AC8 and CaM, Gαs or Gβγ) that were identified in at least one other condition are coloured (CaM = pink, Gαs = green, Gβγ = blue). (**B**) XL-MS analysis of AC8 with Gαs shows that the Gαs N-terminus has multiple contact sites with AC8. (**C**) XL-MS of AC8 with Gαs and CaM shows that CaM and Gαs are in close proximity with each other in a complex. (**D**) XL-MS of AC8 in the presence of Gβγ reveals a set of crosslinks sites between the Gγ and the N-terminus of AC8, and Gβ and the C1b and HD2 domains of AC8. The crosslinking maps in (A–D) were produced with XiNET (Combe et al, 2015). (**E–H**) Sketches of AC8 and the added interactors, with the identified heteromeric crosslinks. Heteromeric crosslinks that were identified in at least one other condition are coloured (CaM = pink, Gαs = green, Gβγ = blue). The AC8-Gβγ crosslinking data facilitate protein docking, which reveals a plausible mode of Gβγ interaction with the AC8 Nt, C1b and C2b. The crosslinking maps in (A–D) were produced with XiNET (Combe et al, 2015). (**I**) Comparison between the AC5-Gβγ cryo-EM-based model (PDB ID: 8SL3) and our XL-MS and docking-based model of AC8-Gβγ complex reveals a remarkable similarity between the Gβγ binding sites in AC5 and AC8 (Yen et al, 2023). (**J**) Residues of AC8 (yellow boxes) participating in crosslinking events with D312 of Gβγ (blue box) were used as docking restraints. The values in Å indicate the distances in the docking model. Docking of the AC8-Gβγ complex shown in (I) and (J) was performed with HADDOCK (van Zundert and Bonvin, 2014).

in these enzymes, stimulating the enzymatic activity. Careful computational and experimental analysis of the influence of CaM on the AC8 catalytic site dynamics will be necessary to establish whether similar modes of regulation apply to the mammalian Ca²⁺/CaM-sensitive ACs.

The combination of two structural proteomics approaches (LiP-MS and XL-MS) allowed us to overcome this inherent limitation and to (i) probe the conformational changes associated with regulator binding, and (ii) assess the candidate interaction sites and distances between AC8 and its regulators, deriving useful structural

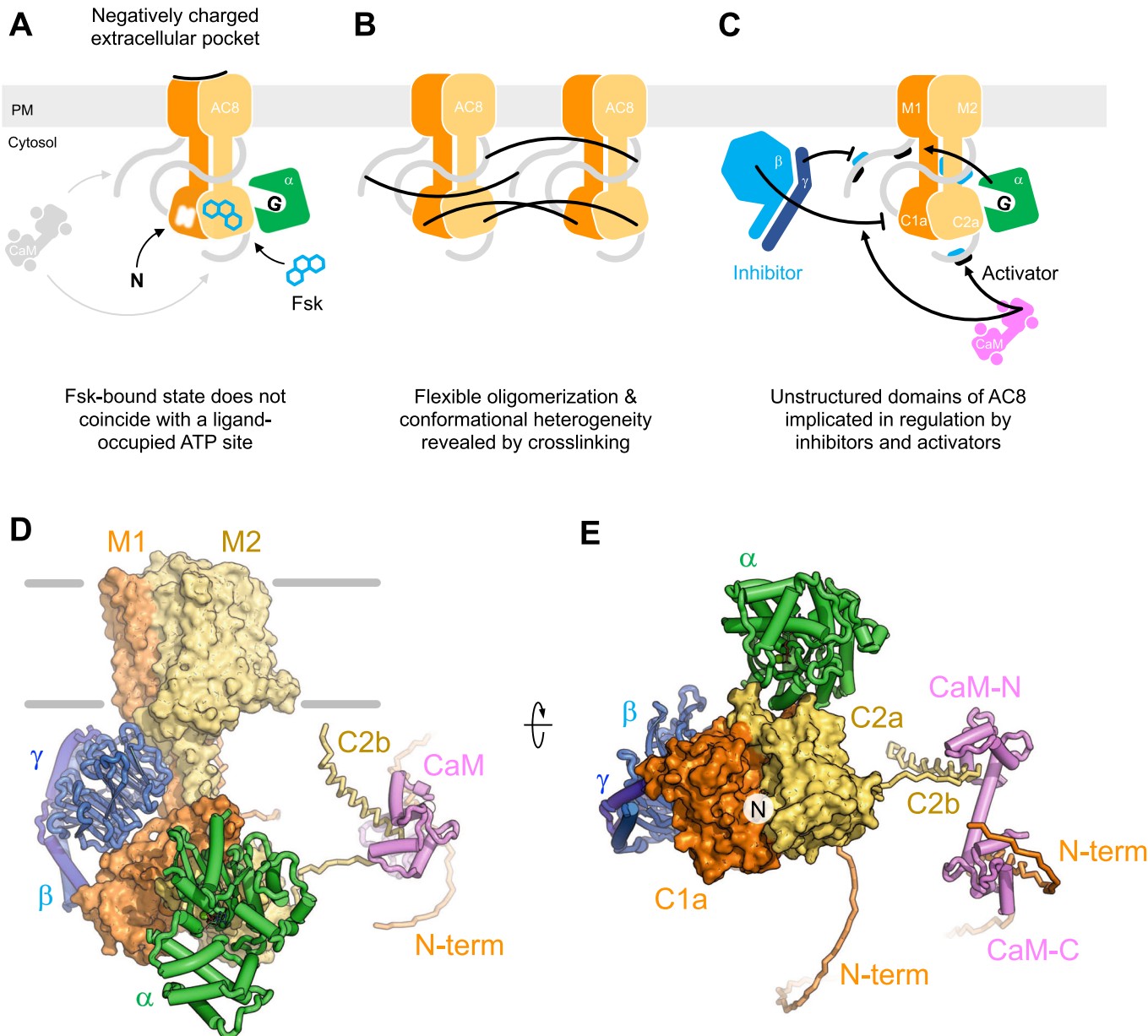

**Figure 7. Insights into the structure of AC8 and interactions with its activators.**

(**A**) Cryo-EM structures of AC8 reveal a negatively charged extracellular pocket (red), and dynamic nucleotide binding in presence of a small molecule activator, forskolin (Fsk). (**B**) XL-MS of purified AC8 provides direct evidence for its oligomerisation. AC8 oligomerisation involves the N-terminus, the C1b domain and the C-terminus. (**C**) XL-MS and LiP-MS experiments show that the flexible N-, C-, and C1b domains of AC8 play vital roles in interactions with and regulation of AC8 by the activators (Gαs and CaM) and inhibitors (Gβγ). (**D, E**) An illustrative model of AC8 and its activating (Gαs and CaM) or inhibitory (Gβγ) interactors, based on cryo-EM (AC8-Gαs-CaM), docking (AC8-Gβγ) and AlphaFold2 (AC8-Gαs-CaM).

information. In addition to detecting the ability of AC8 to form oligomeric assemblies with a great degree of flexibility and/or heterogeneity (Fig. 7B), our data faithfully reproduce the previously observed CaM interactions at sites within the AC8 N-terminus and the C2b domain (Fig. 7C). To further illustrate the different modes of AC8 regulation by distinct binders, we complement our cryo-EM structure with the XL-MS-based docking model of AC8-Gβγ and the AlphaFold2 model of AC8-CaM-Gαs (Fig. EV5). The Alpha-Fold2 model recapitulates the CaM-binding sites in the N-terminus

of AC8 and in the C2b domain, consistent with our data and with the published results (Herbst et al, 2013; Masada et al, 2009) remarkably well. This model helps us to rationalise the absence of a defined EM density for CaM in our cryo-EM maps, as the CaM-binding sites in AC8 are located on highly flexible regions and distant from the ordered part of AC8 upon activation.

Our interpretations rely on the experimental evidence, cryo-EM analysis, and AlphaFold2-based predictions. We believe that our cryo-EM-based reconstruction of the AC8-CaM-Gαs complex

represents a CaM- and Gαs-bound complex wherein CaM and its binding sites cannot be resolved by EM imaging. Nevertheless, it is worth noting that additional effects may affect the appearance of the particles under cryo-EM sample preparation and imaging conditions. For example, a biomolecular complex may prove to be unstable upon application to the cryo-EM grid, or upon exposure to the air-water interface. While we have no evidence that this is the case for the AC8-CaM-Gαs complex, due caution needs to be exercised particularly when inferring the presence of the components which cannot be directly observed by imaging (i.e., CaM and the flexible, unstructured domains of AC8), even despite the available abundant evidence in support of their presence.

The combined illustrative model shown in Fig. 7D,E, along with our experimental results, provides the framework for understanding how Gαs, Gβγ and CaM engage AC8 via multiple interactions at the non-overlapping structured and unstructured regions, leading to activation or inhibition of cAMP production. While structural biology methods such as cryo-EM are not suitable for analysing the interactions of AC modulators with the unstructured regulatory sites of AC8 or other ACs, the MS-based approaches have shown their tremendous utility. The combination of cryo-EM, LiP-MS and XL-MS implemented here provides a powerful tool to study challenging membrane protein complexes that feature structured and unstructured domains.

# Methods

## Chemicals

Detergents, dodecyl-β-maltoside (DDM), glyco-diosgenin (GDN), digitonin, cholesteryl hemisuccinate (CHS) and brain polar lipids were purchased from Anatrace Inc. Calmodulin (from bovine testes, Catalog No: P1431) was purchased from Sigma-Aldrich. All other chemicals were obtained from Sigma-Aldrich (St. Louis, MO, USA), unless indicated otherwise.

## Protein expression and purification

### Adenylyl cyclase AC8

The full-length bovine AC8 (UniProt ID: E1BQ12) was cloned into pACMV-based tetracycline-inducible vector with C-terminal 3C-YFP-TwinStrep fusion tag. The plasmids were transfected into HEK293S GnTI⁻ cells (RRID: CVCL_A785; not authenticated, not checked for mycoplasma) using branched polyethyleneimine (PEI) and a stable monoclonal cell line (HEK293-AC8) capable of expressing AC8 was generated for large scale expression. HEK293-AC8 cells were cultured in Protein Expression Medium (PEM, Gibco) to a cell density of $1.8 \times 10^6$ ml$^{-1}$ and expression was induced by adding 200 μL of 10 mg/mL tetracycline. For purification, the cell pellets were resuspended in 50 mM Tris-HCl pH 8.0, 150 mM NaCl, 10% glycerol supplemented with protease inhibitors (1 mM benzamidine, 1 μg/ml leupeptin, 1 μg/ml aprotinin, 1 μg/ml pepstatin, 1 μg/ml trypsin inhibitor and 1 mM phenylmethylsulfonyl fluoride (PMSF)). Cells were lysed using a Dounce homogeniser and the total cell membranes were collected by ultracentrifugation (Ti45 rotor, $186,000 \times g$ for 40 min at 4 °C). Membranes were flash-frozen and stored at −80 °C until needed the day of experiment. On the day of purification, membranes were thawed, and resuspended

in the 50 mM Tris-HCl pH 8.0, 150 mM NaCl, 10% Glycerol, 1% DDM and 0.02% cholesteryl hemisuccinate (CHS) and solubilised for 1 h at 4 °C. The detergent solubilised lysate was cleared by ultracentrifugation (Ti45 rotor, $186,000 \times g$ for 40 min at 4 °C) to remove insoluble debris. The supernatant was incubated with 4 mL of CNBr-activated Sepharose coupled to an anti-GFP nanobody. After a 60 min incubation at 4 °C the resin was collected in a gravity column and washed with 40 column volumes of 50 mM Tris-HCl pH 8.0, 150 mM NaCl, 0.02% GDN, 5% glycerol. The protein was eluted using cleavage by HRV 3 C protease (1:10 w/w). The eluted protein was concentrated with a 100 kDa cut-off Amicon Ultraspin device (Millipore) and subjected to size-exclusion chromatography (SEC) using a Superpose 6 Increase 10/300 GL column (GE Healthcare) equilibrated in 50 mM Tris-HCl pH 8.0, 150 mM NaCl, 0.02% GDN. The fractions corresponding to purified AC8 (elution volume, 13–16 mL) were collected. Finally, SEC fractions were concentrated and used directly for cryo-EM grids preparation or flash-frozen in 10% glycerol and stored at −80 °C. The protein purity was assessed using 4–20% SDS-PAGE (BIO-RAD) and visualised by standard Coomassie brilliant blue staining technique. Protein concentration was determined by absorbance at 280 nm using ε = 126,975 M$^{-1}$ cm$^{-1}$.

### G protein αs subunit

The procedure for expression and purification of the bovine Gαs (UniProt ID P04896-1, with a C-terminal 8xHis-tag) was similar to the one previously described (Qi et al, 2022, 2019). The standard Bac-to-Bac baculovirus expression system was used (Invitrogen): High Five (Hi5) insect cells were cultured in suspension to a cell density of $1.5 \times 10^6$ ml$^{-1}$ and 1–2% P2 virus was added to infect the cells for protein expression. The cells were harvested after 72 h. For protein purification, the cells were resuspended in buffer A, lysed using a Dounce homogeniser and solubilised by adding 1% DDM for 1 h. The lysate was clarified by ultracentrifugation (Ti45 rotor, $186,000 \times g$ for 40 min at 4 °C). The supernatant was incubated with 1 ml Ni-NTA resin for 30 min. The resin was washed with 50 mM Tris-HCl pH 8.0, 150 mM NaCl, 0.02% DDM, and 20 mM imidazole. A second wash followed this with 50 mM Tris-HCl pH 8.0, 150 mM NaCl, 0.02% GDN, 5% glycerol, 40 mM imidazole. The protein was eluted with 50 mM Tris-HCl pH 8.0, 150 mM NaCl, 0.02% GDN, 5% glycerol, 250 mM imidazole. The eluted protein was concentrated with a 30-kDa cut-off Amicon Ultraspin device (Millipore) and injected onto a Superose 6 Increase 10/300 GL column (GE Healthcare) pre-equilibrated in 50 mM Tris-HCl pH 8.0, 150 mM NaCl, 0.02% GDN. The peak fractions (elution volume, 14–16 mL) were concentrated, snap-frozen in liquid nitrogen in aliquots, and stored at −80 °C until the day of the experiment. Protein concentration was determined by absorbance at 280 nm using ε = 43,360 M$^{-1}$ cm$^{-1}$.

### G protein beta gamma subunit

Gβγ was expressed from insect cells (Hi5 cells) using the flashBAC system (Hitchman et al, 2012). Hi5 insect cells were cultured in serum free SF-4 medium to a cell density of $1.5 \times 10^6$ ml$^{-1}$ and the expression of protein was induced by adding 1–2% P2 virus. After 48 h cells were harvested by centrifugation at $1000 \times g$ for 15 min. Cell pellets were resuspended in buffer A, lysed using a Dounce homogeniser and the total cell membranes were collected by ultracentrifugation (Ti45 rotor, $186,000 \times g$ for 40 min at 4 °C). The

supernatant was discarded and the membrane pellet was resuspended in 50 mM Tris-HCl (pH 8), 200 mM NaCl, 1% sodium cholate, 10% glycerol, 25 mM imidazole and left rotating for 1 h at 4 °C. The suspension was clarified using ultracentrifugation at 40,000 rpm (186,000 × g; Beckman Ti45 rotor) for 40 min at 4 °C and the supernatant was incubated with 2 mL HisPure cobalt resin (Thermo Fisher Scientific) per 1 L original culture for 1 h at 4 °C under constant agitation. The bead/sample suspension was loaded onto a gravity column and washed with 40 column volumes of 50 mM Tris pH 8.0, 150 mM NaCl, 10% glycerol, 25 mM imidazole and 0.1% digitonin, followed by washing with 40 column volumes of 50 mM Tris pH 8.0, 150 mM NaCl, 10% glycerol, 50 mM imidazole and 0.1% digitonin). Gβγ was eluted with 50 mM Tris pH 8.0, 150 mM NaCl, 10% glycerol, 400 mM imidazole and 0.1% digitonin and the His-tag was cleaved over night with HRV 3 C protease (amount: 1:50 eluted protein:HRV 3 C). The flow-through was collected and desalted with buffer (50 mM Tris pH 8.0, 150 mM NaCl, 10% glycerol and 0.1% digitonin) using a GE PD-10 desalting column. The desalted protein was then passed through Ni-NTA resin to remove HRV 3 C protease, concentrated to 1 mL and injected onto a Superose 6 Increase 10/300 GL column (GE Healthcare) equilibrated in 50 mM Tris-HCl, 150 mM NaCl, 0.02% GDN, pH 8.0. Fractions (elution volume, 13–16 mL) containing Gβγ were further concentrated, flash frozen in liquid nitrogen with the addition of 10% glycerol and stored at −80 °C until further use.

### GFP-nanobody

The expression and purification of anti-GFP nanobody was carried out as previously described (Kubala et al, 2010). In short, the anti-GFP nanobody plasmid was transformed into *E. coli* BL21 (DE3) and grown in Luria Broth (LB) medium supplemented with 50 µg/mL ampicillin at 37 °C. Once the optical density at 600 nm (OD600) of the bacterial culture reached 0.5, the expression of protein was induced by adding 0.5 mM isopropyl β-D-1-thiogalactopyranoside (IPTG) followed by overnight incubation at the 20 °C. Bacterial cultures were harvested at 4000 × g for 20 min at 4 °C, and pellets were frozen in liquid N2 and stored at −80 °C. Frozen pellets were thawed on ice, re-suspended in 25 mM HEPES pH 8.0, 150 mM NaCl, 10 mM imidazole, 1 mM PMSF, 10 µg/mL DNase I. The cells were lysed by sonication and the cell lysate was centrifuged for 30 min at 20,000 × g. Clarified lysate was then incubated with Ni-NTA resin (1–2 mL bed volume of resin per 1 L of culture) for 30–40 min, and subsequently washed with 20 CV of 25 mM HEPES pH 8.0, 150 mM NaCl, 50 mM imidazole, the protein was eluted with 5 CV of the same buffer with 250 mM imidazole. The elution was concentrated with 10 kDa cut-off Amicon concentrator and loaded on a Superdex 75 16/600 GL column (GE Healthcare) in 25 mM HEPES pH 8.0, 150 mM NaCl. SEC fraction corresponding to GFP-nanobody was pooled and flash frozen in liquid nitrogen, and stored at −80 °C. Protein concentration was determined by absorbance at 280 nm using $\varepsilon = 27055\ M^{-1}\ cm^{-1}$.

### Membrane scaffold protein

The expression and purification of Membrane Scaffold Protein MSP1E3D1 was carried out as previously described (Ritchie et al, 2009). In brief, the MSP1E3D1 plasmid was transformed into E. coli BL21 (DE3) and grown at 37 °C in Terrific Broth (TB) media. The expression of protein was induced with 1 mM IPTG at OD600

of ~2–3 and incubation for 3 h. After harvesting by centrifugation, cell pellets were re-suspended in lysis buffer (50 mM Tris-HCl pH 8.0, 200 mM NaCl, 25 mM imidazole, 1% Triton-X100, 1 mM PMSF and 10 µg/ml DNaseI) and cells were lysed by sonication. The clarified lysate after centrifugation (20,000 × g for 30 min) was incubated with Ni-NTA resin for 30 min, and subsequently step washed with 10 CV of buffer (50 mM Tris-HCl pH 8.0, 150 mM NaCl, 25 mM imidazole, 1% Triton-X100), 5 CV of buffer (50 mM Tris-HCl pH 8.0, 150 mM NaCl, 25 mM imidazole, 2% Sodium Cholate), and 5 column volumes of buffer (50 mM Tris-HCl pH 8.0, 150 mM NaCl, 50 mM imidazole). Finally, the protein was eluted in 50 mM Tris-HCl pH 8.0, 150 mM NaCl, 350 mM imidazole. Elution fractions containing MSP1E3D1 were pooled, desalted in buffer (20 mM Tris-HCl pH 8.0, 200 mM NaCl) and flash frozen in liquid nitrogen, and stored at −80 °C. Protein concentration was determined by absorbance at 280 nm using $\varepsilon = 29,400\ M^{-1}\ cm^{-1}$.

### Calmodulin-YFP

The HEK293F cells grown in suspension at a cell density of 1.5 million cells/ml of culture were transfected with an expression plasmid containing the CaM-YFP-twinStrep (referred to here and throughout the manuscript as CaM-YFP) construct (1 mg DNA per 1 L of cells, using linear PEI at a DNA to PEI ratio of 1:3). The cells were incubated in a shaker for 72 h at 37 °C, at 5% CO2 and 120 rpm, after which the cell pellets were harvested by centrifugation at 2000 rpm at 4 °C for 20 min using a 6 × 1000 mL swinging bucket rotor (Sorvall BIOS 16 Centrifuge, Thermo Scientific). The collected cell pellets were frozen and stored at −80 °C until the day of purification. To purify CaM-YFP, the cell pellets corresponding to 1 L of culture were thawed, resuspended in a cold buffer (50 mM Tris-HCl, pH 7.5, 200 mM NaCl, 2 mM EGTA, 1 µg/µL DNase I, 1x Roche Complete EDTA-free protease inhibitor cocktail) and lysed using a Dounce homogeniser. The lysate was clarified by ultracentrifugation (Ti45 rotor, 186,000 × g for 40 min at 4 °C), and the supernatant was incubated with 4 ml Strep-Tactin Superflow resin for 30 min at 4 °C with rotation. After incubation, the sample was applied to a gravity column and washed with 40 column volumes of buffer (50 mM Tris-HCl, pH 7.5, 200 mM NaCl, 2 mM EGTA). The protein was eluted with 5 column volumes of buffer (50 mM Tris-HCl, pH 7.5, 200 mM NaCl, 2 mM EGTA) containing 5 mM desthiobiotin. The sample was concentrated with a 10 kDa cut-off AmiconUltra concentrator to a final volume of 1 ml, and further purified by size-exclusion chromatography using a Superdex 200 10/300 GL column pre-equilibrated with a buffer (50 mM Tris-HCl, pH 7.5, 200 mM NaCl, 2 mM EGTA) containing additionally 10% glycerol. The fractions corresponding to CaM-YFP were pooled, concentrated, aliquoted, flash frozen in liquid nitrogen, and stored at −80 °C until the day of experiment.

## Parallel reaction monitoring (PRM)-based quantification of AC8 and CaM

### Sample preparation

Two aliquots of purified AC8 (10 µg) at a concentration of 0.23 mg/mL were digested with protiFi S-Trap™ micro columns according to the manufacturer's protocol. After drying in a vacuum centrifuge, the peptide mixture was resuspended in 50 µL 5% acetonitrile (ACN), 0.1% formic acid (FA) in water.

Heavy labelled AQUA peptides (4 peptides for AC8, 3 peptides for CaM; listed in Appendix Table S1) were custom synthesised and ordered from ThermoFisher Scientific. A bovine serum albumin (BSA) background mix containing 100 fmol/µL BSA peptides, iRT peptides (Biognosys), 5% ACN and 0.1% FA in water was prepared. The AQUA peptides were mixed and diluted in 1:10 steps to obtain the calibration curve samples. Briefly, for the 500 fmol/µL sample 2 µL of each peptide stock (5 pmol/µL) were mixed and 6 µL of the BSA background mix were added (20 µL final volume). This sample was sequentially diluted in 1:10 dilution steps (2 µL sample + 18 µL BSA background) to obtain samples containing 50 fmol/µL, 5 fmol/µL, 500 amol/µL, 50 amol/µL, 5 amol/µL and 500 zmol/µL of each peptide.

Purified samples were prepared by diluting the samples 1:4 in 5% ACN, 0.1% FA, with the addition of AC8 AQUA peptides at ca. 500 fmol/µL and CaM AQUA peptides at ca. 500 amol/µL, iRT peptides and BSA background.

### LC-MS/MS – PRM data acquisition

The samples were analysed on an Exploris 480 mass spectrometer (Thermo Fisher Scientific) connected to a Vanquish Neo (Thermo Fisher Scientific) UHPLC. In separate runs, 0.5 µL and 1 µL of the samples were injected and peptides were separated on a 40 cm × 0.75 µm (inner diameter) column in-house packed with 1.9 µm C18 beads (Dr. Maisch Reprosil-Pur 12) over a 30 min linear gradient from 7–35% B (A: 0.1% FA, B: 80% ACN, 0.1% FA) at 300 nL/min, at 50 °C. MS1 spectra were acquired within a scan range between 150–2000 $m/z$ at an Orbitrap resolution of 30,000. RF lens was set to 50%, the AGC target was set to "Standard". The selected precursors (including iRT peptides) and settings are listed in Appendix Table S1.

### Data analysis

The acquired data was analysed in Skyline (v. 23.0.9.187), exported and further interpreted in R using the R packages tidyverse and (Wickham, 2019) data.table (Barrett et al, 2024). The retention times of the detected iRT peptides were checked for consistency. To produce the calibration curve, the calculated areas of the five highest ranked fragments of heavy peptides were summed up for each sample. After assessing linearity of the calibration curve (LOQ) (Appendix Fig. S2), the two lowest concentration points (500 zmol/µL and 5 amol/µL) were removed. A linear model (lm) was fit onto the log10 transformed, summed up area values at the specific log10 transformed concentration points (log10(area) ~ log10(concentration)) and the slope and intercept were calculated for each heavy labelled AQUA peptide.

To assess AC8 and CaM sample concentrations and their stoichiometry the five highest ranked fragments of light peptides were summed up for each sample. The slope and intercept values obtained from the respective calibration curves were used to calculate the absolute concentrations of peptides and proteins (average of peptide intensity). Peptide EAFSLFDK (CaM) was excluded from the calculations, since the measurement points were outside of the range of linearity. The average injected protein amount of AC8 for 0.5 µL injection volume was calculated to be 60.69 ± 11.39 fmol (mean ± standard error of the mean), for 1 µL 113.58 ± 21.03 fmol. The average amount of CaM for 0.5 µL injection volume was 0.17 ± 0.07 fmol, for 1 µL 0.33 ± 0.12 fmol. The ratio of CaM:AC8 in the purified protein samples was in the

range of 1:200–1:720. CaM can therefore be estimated to be copurified with AC8 in substoichiometric amounts at an approximate ratio of 1:500 (CaM:AC8).

### Reconstitution of AC8 into nanodiscs

Reconstitution of AC8 in MSP1E3D1 nanodiscs was performed using freshly purified protein. In brief, 2.5 mg of brain polar lipid (BPL) dissolved in 100 µL of chloroform was dried under a stream of nitrogen. The dried film of lipid was mixed with 300 µL of 3% DDM and the lipid-detergent mixture was then sonicated using bath sonicator (Bandelin SONOREX™SUPER, Germany) until the mixture turned translucent. Detergent-solubilised BPL extract was added to freshly purified AC8 at a molar ratio of 1:150 and incubated for 30 min at room temperature with gentle rotation. MSP1E3D1 was added to the protein-lipid mixture and incubated for additional 30 min. Concentrations of AC8 for nanodisc reconstitution were in the 2–3 µM range, with a molar ratio of AC8 to MSP1E3D1 to lipid of 1:2:150. Following the incubation period, nanodisc formation was triggered by adding 150 mg of wet Bio-beads (washed with 100% methanol and with Milli-Q water). This final reconstitution mixture was incubated at 4 °C for 16 h (overnight) with gentle mixing. The supernatant was cleared of the beads by letting the beads settle and removing liquid carefully with a pipette. Sample was spun for 10 min at 25,000 × g using a benchtop Eppendorf centrifuge before loading onto a Superose 6 Increase 10/300GL column equilibrated in 20 mM Tris-HCl pH 8.0, 150 mM NaCl. The peak fractions corresponding to AC8 in MSP1E3D1 (elution volume, 13–15.2 mL) were collected, concentrated with a 100 kDa cutoff Amicon concentrator and used for cryo-EM grid preparation.

### Adenylyl cyclase activity assays

The cAMP accumulation assay was performed using protocol adapted from Alvarez and Daniels (Alvarez and Daniels, 1990; Qi et al, 2022, 2019). Briefly, the purified AC8 was diluted to 20–30 nM in buffer A (50 mM Tris PH 7.5, NaCl, 0.02% GDN) for all the experiments. Similarly, AC8 complexes with varying concentration of CaM, Gαs, Gβγ and forskolin were prepared. The enzymatic reaction was then initiated by mixing 100 µL of protein solution with 100 µL of 2X reaction mixture containing 4 mM $MgCl_2$, 10 mM $MnCl_2$, 0.2 mM ATP, and 20 nM of $H^3$-ATP (PerkinElmer). The reaction mixture was then incubated at 30 °C for 30 min. After incubation, the reaction was then quenched by the addition of 20 µL of 2.2 M HCl for 4 min at 95 °C. The quenched reaction mixture was immediately cooled on ice, followed by loading onto gravity columns packed with 1.3 g of packed aluminium oxide. cAMP which binds less strongly than ATP, was eluted with 4 ml of 0.2 M ammonium acetate. Before counting, 12 mL of scintillation liquid (LabLogic) was added, and the final solution was brought to counting on a Packard 2250 CA Tri Carb liquid scintillation counter.

### CaM binding assays

AC8-CaM interaction was analysed using fluorescence size exclusion chromatography (FSEC). For this purpose, the purified CaM-YFP was used as a fluorescent probe and binding of CaM to

AC8 was detected based on the emergence of YFP fluorescence at the position of the eluted AC8 peak in an analytical SEC experiment. In brief, purified tag-free AC8 (100 nM final concentration) was mixed with various amounts of CaM-YFP (at a range of concentrations, from 3.75 to 1000 nM) in a final volume of 100 µl. The mixtures were incubated on ice for 15 min, in the presence of $CaCl_2$ (1 mM) or EGTA (1 mM). After the incubation, each sample was injected onto the Agilent SEC5-300 column (flow 0.3 ml/min, excitation 488 nm, emission 510 nm) pre-equilibrated in 50 mM Tris HCl, pH 8.0, 150 mM NaCl, 0.02% GDN and 1 mM $CaCl_2$ or 1 mM EGTA. The total peak area (elution volume, 1.2–1.62 ml) and peak intensity (at 1.47 ml) were quantified for each experiment. The binding affinity of AC8 and CaM was evaluated using the one-site saturation binding model in GraphPad Prism 8. Co-localisation of CaM-YFP with AC8 in the complex peak was validated by analysing the full AC8-CaM-YFP complex (1:2 ratio) by SEC, using Superose 6 increase column, followed by SDS-PAGE, in-gel fluorescence (using BIO-RAD ChemiDoc™ MP Imaging System device) and Coomassie blue staining.

## Cryo-EM sample preparation and data collection

A typical sample of AC8-CaM-Gαs complex was prepared by mixing concentrated AC8 (~6 mg/mL) with 1.2-fold molar excess of GTPγS-activated Gαs protein, twofold excess of calmodulin (purchased from Sigma-Aldrich, Catalog No: P1431) and incubated with 0.5 mM MANT-GTP, 0.1 mM GTPγS, 1 mM $CaCl_2$, 2 mM $MgCl_2$, and 5 mM $MnCl_2$. The final concentration of AC8 in the sample was ~5 mg/ml. The final Quantifoil 1.2/1.3 200-mesh grids were briefly glow discharged in a PELCO easiGlow (Ted Pella) glow discharge cleaning system for 25 s at 30 mA in air. A 3.5 µL sample was deposited on the carbon side of the grid and blotted for 3 s (blot force of 20) inside the chamber of a Vitrobot Mark IV (Thermo Fisher Scientific) with 100% humidity and at 4 °C. The grid was then plunged into liquid ethane, and the grids were stored in liquid nitrogen. A total of three datasets with 3618, 5178, and 7149 movies were collected using EPU on a 300 kV Titan Krios (Thermo Fisher Scientific) equipped with a Gatan K3 direct electron detector and a Gatan Quantum-LS GIF at ScopeM, ETH Zurich. All movies were acquired in super-resolution mode with a defocus range of −0.5 to −3 µm and a final calibrated pixel size of 0.33 Å. The total dose per movie was 60, 56 and 49 e⁻/Å² for datasets 1, 2, and 3, respectively.

For AC8-Gαs-Ca²⁺/CaM complex in lipid nanodiscs, the AC8 reconstituted in lipid nanodiscs at a concentration of 4 mg/ml was incubated with 1.2-fold molar excess of GTPγS -activated Gαs protein, twofold excess of calmodulin 1 mM ATPαs, 0.1 mM GTPγS, 1 mM $CaCl_2$, 2 mM $MgCl_2$, and 5 mM $MnCl_2$ for 10 min at 4 °C. A small aliquot of the sample (3.5 µl) was applied to the glow-discharged Quantifoil R1.2/1.3 200-mesh grid. The grid was blotted for 3 s and plunge-frozen in liquid ethane using a Vitrobot Mark IV (Thermo Fisher Scientific). The grids were transferred to and stored in liquid nitrogen for subsequent cryo-EM data collection. A total of three datasets with 2007, 8022, and 11,291 movies were collected using EPU on a 300 kV Titan Krios (Thermo Fisher Scientific) equipped with a Gatan K3 direct electron detector and a Gatan Quantum-LS GIF at ScopeM, ETH Zurich. All movies were acquired in super-resolution mode with a defocus range of −0.5 to −3 µm and twofold binned with final calibrated pixel size of 0.66 Å. Movies of 40 frames were collected with a total dose of 55, 55, and 56.4 e⁻/Å² for datasets 1, 2, and 3, respectively.

## Cryo-EM data analysis and model building

The data processing of AC8-CaM-Gαs was performed in Relion (3.0 and 3.1-beta versions) (Zivanov et al, 2018). All movie stacks were motion corrected using MotionCorr 1.1.0 (Zheng et al, 2017) and binned twofold to yield a magnified pixel size of 0.66. All micrographs were CTF corrected using GCTF (Zhang, 2016). Initially, around 1017 particles were manually picked to generate 2D template for autopicking. Selected 2D classes from manually picked particles were used to autopick particles from all micrographs in dataset 1. After multiple rounds of 2D and 3D classification jobs, the best 3D class from first dataset was used to repick the particle with 3D projections for all three of AC8-CaM-Gαs complex datasets. 5.72 million particles were picked from three datasets. After multiple rounds of 2D and 3D clarifications, a set of 615,431 particles was selected for masked 3D refinement (using masks including or excluding the nanodisc density), resulting in a 3D reconstruction at 4.5 Å. The refined particles were subjected to another round of 3D classification without alignment and with masking of the detergent. The particles from the best 3D class were subjected to several iterative cycles of 3D refinement, CTF refinement and particle polishing, yielding a final post-processed density at 3.55 Å resolution. Local resolution maps were calculated by ResMap implemented in Relion 3.1 (Kucukelbir et al, 2014; Zivanov et al, 2018). The detailed steps of cryo-EM processing are depicted in Appendix Fig. S5.

The data processing of AC8-CaM-Gαs in nanodisc was performed in Relion 3.1.3. All movie stacks were motion corrected using MotionCorr 1.1.0 (Zheng et al, 2017) and CTF corrected using Gctf (Zhang, 2016). The rest of image processing was similar to that of AC8-CaM-Gαs sample in detergent. In brief, a total of 6,127,395 particles were autopicked from three datasets using 3D structured obtained in detergent conditions. After multiple rounds of 2D and 3D clarifications, a set of 33,1737 particles was selected for masked 3D refinement (using masks including or excluding the nanodisc density), resulting in a 3D reconstruction at 3.97 Å resolution. The refined particles were subjected to several iterative cycles of 3D nonalignment classification, 3D refinement, CTF refinement and particle polishing, yielding a small improvement in quality of map without changes in resolution. Local resolution maps were calculated by ResMap implemented in Relion 4.0 (Kimanius et al, 2021; Kucukelbir et al, 2014). The detailed steps of cryo-EM processing are depicted in Appendix Fig. S7.

## Limited proteolysis-mass spectrometry (LiP-MS)

### Membrane suspension preparation

A pellet of 2 L induced HEK 293S GnTI⁻ cells was thawed on ice and resuspended in 50 mL 100 mM HEPES-KOH (pH 7.4), 150 mM KCl, 1 mM $MgCl_2$ (=LiP buffer) with 2 Roche cOmplete protease inhibitor cocktail tablets and DNAse A at a final concentration of 10 µg/mL. The suspension was homogenised using a dounce homogeniser (20 strokes). The sample was further centrifuged at $1000 \times g$ for 10 min at 4 °C, the supernatant was divided in half, filled up to 50 mL and centrifuged at 35,000 rpm

(142,000 × g; Beckman Ti45 rotor) for 40 min at 4 °C. Both pellets were resuspended in a total of 15 mL LiP buffer and homogenised with a Dounce homogeniser (20 strokes). The sample was aliquoted, snap frozen in liquid nitrogen and stored at −80 °C until used.

The protein concentration of the membrane suspensions was determined using the BCA assay (Pierce BCA Protein Assay Kit).

### LiP-MS Gβγ titration

The previously prepared membrane suspensions were thawed on ice and diluted to 2 µg/µL in LiP buffer with 1 mM $CaCl_2$, 1 mM $MnCl_2$ and 100 µM GTPγS. Purified Gβγ was diluted in LiP buffer with 1 mM $CaCl_2$, 1 mM $MnCl_2$ and 100 µM GTPγS and 0.1% digitonin. All experiments were conducted in quadruplicates. 2 µL of diluted Gβγ at different concentrations (1.5, 1, 0.5, 0.25, 0.05 and 0.005 µg/µL) and a buffer blank was spiked into 50 µL of AC8 membrane suspension and incubated for 10 min at 25 °C in a Thermocycler. 5 µL of proteinase K from *Tritirachium album* (Sigma Aldrich) at a concentration of 0.2 µg/µL were added to the samples and incubated for 5 min at 25 °C. Proteinase K was inactivated by heating the samples to 99 °C for 5 min, cooling them at 4 °C for 5 min and further addition of 5% sodium deoxycholate (final concentration).

In addition, tryptic controls (TC) were produced for the buffer blank treated and the samples treated with the highest amount of Gβγ in quadruplicates: to 50 µL of membrane suspension 2 µL of buffer blank or 2 µL of Gβγ (1.5 µg/µL) were spiked and incubated for 10 min. 5 µL of water (instead of proteinase K) were added incubated for 5 min at 25 °C. The samples were then treated the same way as the LiP samples.

### LiP-MS Gαs titration

The experiment was conducted in a similar way to that with the Gβγ. Briefly, samples were diluted in LiP buffer with 1 mM $CaCl_2$, 1 mM $MnCl_2$ and 100 uM GTPγS and Gαs was titrated in a range of 0–3 µg per sample in the same steps as used for the Gβγ titration.

### Tryptic digest

All LiP and TC samples were subjected to a tryptic digest. Briefly, disulfide bonds were reduced with 5 mM tris(2-carboxyethyl) phosphine-hydrochloride for 40 min at 37 °C with slight agitation. Reduced cysteines were alkylated with 40 mM iodoacetamide at room temperature in the dark with slight agitation. The samples were diluted to a final sodium deoxycholate concentration of 1% with 100 mM ammonium bicarbonate. 1 µL of Lysyl endopeptidase LysC (1 µg/µL; IGZ Instruments) and 2 µL of sequencing grade trypsin (0.5 µg/µL; Promega) were added and the samples were incubated overnight at 37 °C with slight agitation. The next day, the digest was stopped by adding formic acid to a final concentration of 2%. The precipitated sodium deoxycholate was removed by filtering through a 96-well Corning 2 µM polyvinylidene fluoride (PVDF) plate and the samples were desalted on a 96-well MacroSpin plate (The Nest Group). The peptides were eluted with 80% acetonitrile, 0.1% formic acid and dried in a vacuum centrifuge. The samples were stored at −20 °C until further use.

### LC-MS/MS

The samples were reconstituted in 5% ACN, 0.1% FA with the addition of iRT peptides (Biognosys). For library generation,

pooled samples of all replicates of one condition were measured in DDA (data dependent acquisition). The unpooled samples were measured in DIA (data independent acquisition). Gβγ titration data was acquired on an Orbitrap Fusion Tribrid mass spectrometer (Thermo Fisher Scientific), equipped with a nanoelectrospray source and an Easy-nLC 1200 nanoflow LC system (Thermo Fisher Scientific). Peptides were separated on a 40 cm × 0.75 i.d. column packed in-house with 1.9 um C18 beads (Dr. Maisch Reprosil-Pur 120) using a linear gradient from 3 to 35% B (A: 0.1% FA, B: 95% ACN, 0.1% FA) over 120 min at 300 nL/min. The column was heated to 50 °C. DIA samples were acquired with a 41 window (1 *m/z* overlap) method. For DDA measurements the Orbitrap resolution was set to 120,000, the scan range was between 350 and 2000 *m/z*. Maximum injection time was 50 ms, the normalised AGC target was set to 200%, RF lens was set to 60%. Exclusion time was 30 s, charge states between +2 and +6 were analysed. The fragmentation method was higher energy collisional dissociation (HCD) with a collision energy of 30% and a scan range of 150–2000 *m/z*. For DIA measurements a survey MS1 scan was acquired at an Orbitrap resolution of 120,000 with a scan range between 350 and 1400 *m/z* and a maximum injection time of 100 ms. The normalised AGC target was at 200%, RF lens at 30%. Consecutively, 41 variably sised MS2 windows with 1 *m/z* overlap were acquired. The Orbitrap resolution was set to 30,000, the HCD collision energy to 30%. The normalised AGC target was set to 100%.

The Gαs titration samples were acquired on an Orbitrap Eclipse Tribrid mass spectrometer (Thermo Fisher Scientific). For DDA measurements the Orbitrap resolution was set to 120,000, the scan range was between 350 and 1400 *m/z*. Maximum injection time was 54 s, the normalised AGC target was set to 200%. RF lens was set to 30% and charge states between +2 and +7 were analysed. The exclusion time was set to 20 s. The fragmentation method was HCD with a collision energy of 30%. The Orbitrap resolution was set to 30,000, the scan range was between 150 and 2000 *m/z*. Maximum injection time was 54 ms, the normalised AGC target was set to 200%. For DIA measurements a survey MS1 scan was acquired with the Orbitrap resolution set to 120,000 and a scan range between 350 and 1400 *m/z*. The injection time was set to 100 ms, the normalised AGC target to 200%, RF lens was at 30%. The DIA window setup was the same as mentioned above for the Orbitrap Fusion Tribrid mass spectrometer method.

### Data analysis

The acquired raw data was analysed using Spectronaut v. 15.4 (Bruderer et al, 2015). Tryptic control data and LiP data was searched separately. A library was produced from searches of DDA data. Default search parameters were slightly altered: a minimum peptide length of 5 amino acids was applied and single hits were excluded, missing values were not imputed. For LiP-MS data, protease specificity was set to "semi-specific". Data was exported from Spectronaut and further analysed in R. Quality control, data analysis and interpretation was conducted using the R packages protti (Quast et al, 2022), tidyverse (Wickham, 2019), data.table (Dowle and Srinivasan, 2022). The tryptic control data was checked for AC8 protein abundance changes and four-parameter dose response curves were fit onto all AC8 peptides detected in the LiP samples using an implementation of the drc (Ritz et al, 2015) R package. Pearson correlation was calculated and peptides with a

Pearson correlation r of >0.85 were considered significant. Peptides for which only one concentration point was different to the rest were not considered significant.

## Crosslinking-mass spectrometry (XL-MS)

AC8, Gαs and Gβγ were concentrated and the purification buffer was exchanged for 50 mM HEPES pH 7.5, 150 mM NaCl, 0.02% GDN, 0.5 mM CaCl$_2$, 1 mM MgCl$_2$, 2.5 mM MnCl$_2$, 100 μM GTPγS. AC8 crosslinking was performed with 60 μg protein, for the other conditions, 30 μg AC8 was used. G proteins were incubated with 100 μM GTPγS for 10 min at 25 °C prior to crosslinking. The interactor reactions were conducted at molar rations of AC8: Gαs 1:1.5, AC8:Gβγ 1:3, AC8:CaM 1:2, AC8:CaM: Gαs 1:2:1.5. All samples were incubated for 30 min at 25 °C.

Primary amine crosslinking was performed with 1 mM DSS (disuccinimidyl suberate) (11.4 Å linker length (Merkley et al, 2014))at a 1:1 ratio of DSS-d$_0$ and DSS-d$_{12}$ (Creative Molecules; 25 mM stock in dimethyl formamide). Samples were incubated at 25 °C for 1 h at 300 rpm. The crosslinking reaction was quenched with 50 mM ammonium bicarbonate.

Carboxyl group and primary amino group crosslinking was performed with 44 mM PDH (pimelic dihydrazyde)-d$_0$ (ABCR) mixed at a 1:1 ratio with PDH-d$_8$ (Sigma-Aldrich) (12.3 Å linker length (Leitner et al, 2014a)) and DMTMM (4-(4,6-dimethoxy-1,3,5-triazin-2-yl)-4-methylmoorpholinium chloride). Samples were incubated at 25 °C for 1 h at 300 rpm. To quench the reaction, the samples were desalted using Zeba Spin Desalting columns (Thermo Fisher Scientific). An aliquot of each crosslinked and non-crosslinked samples was analysed with SDS-PAGE (Appendix Fig. S9).

All crosslinked samples were dried in a vacuum centrifuge before further processing. The dried samples were resuspended in 50 μL 8 M urea, disulfide bonds were reduced with 2.5 mM tris(2-carboxyethyl) phosphine hydrochloride for 30 min at 37 °C. Free thiol groups were alkylated with 5 mM iodoacetamide for 30 min at room temperature in the dark. 25 μL of 150 mM ammonium bicarbonate were added to the samples, followed by the addition of 0.6 μg LysC (1:100 enzyme:protein). After 2 h incubation at 37 °C 320 μL of 50 mM ammonium bicarbonate were added to the samples. Trypsin (Promega) was added at an enzyme to protein ratio of 1:50. Samples were incubated overnight at 37 °C with slight agitation. Formic acid was added to a concentration of 2% and the samples were desalted with solid-phase extraction (MicroSpin Columns 5–60 μg capacity). The samples were evaporated to dryness and resuspended in 30% ACN, 0.1% trifluoroacetic acid (TFA).

The samples were further fractionated using size-exclusion chromatography (Superdex 30 Increase column 300 × 3.2 mm, GE Healthcare) in 30% ACN, 0.1% TFA at a flow rate of 50 μL/min. Four fractions were collected for each sample and dried in a vacuum centrifuge.

### LC-MS/MS

Ten percent of each SEC fraction—except for the highest molecular weight fractions of AC8-CaM (PDH/DMTMM) and AC8-CaM-Gαs (PDH/DMTMM) (2.5%)—were injected onto an Easy nLC-1200 HPLC (Thermo Fisher Scientific) system coupled to an Orbitrap Fusion Lumos mass spectrometer (Thermo Fisher Scientific). Peptides were separated on an Acclaim PepMap RSLC C18 column (250 mm × 75 μm; Thermo Fisher Scientific) with a

gradient from 11–40% B (A: 2% ACN, 0.15% FA; B: 80% ACN, 0.15% FA) in 60 min, at a flow rate of 300 nL/min.

All samples were acquired in DDA with a cycle time of 3 s and a dynamic exclusion of 30 s. MS1 Orbitrap resolution was set to 120,000, charge states between +3 and +7 were selected in a range of 350–1500 $m/z$ and fragmented with collision-induced dissociation in a linear ion trap at a collision energy of 35%. Fragment ions were detected in in the ion trap.

In addition, PDH/DMTMM samples were measured in duplicates with a method similar to the method above but with fragment detection in the Orbitrap at a resolution of 30,000.

### Data analysis

To determine which proteins were present in the sample, the smallest molecular weight fractions of DSS crosslinked samples from each condition were searched with Spectromine v.3 (Biognosys) against a database of human, bovine, *Trichoplusia ni* and target proteins, as well as common contaminants (used from MaxQuant (Cox & Mann, 2008)). Identified proteins with >5% LFQ intensity (compared to target proteins) were selected for the initial FASTA files. The initial FASTA files were used for crosslink identification with xQuest v. 2.1.5 (Leitner et al, 2014b). The databases were reversed and shuffled. Trypsin was set as the digestion enzyme, maximum number of missed cleavages 2, carbamidomethylation on cysteines was set as a fixed modification, oxidation of methionines was set as a variable modification, peptide lengths were set to 4–40 amino acids. MS1 error tolerance was set to 15 ppm, MS2 tolerance was set to 0.2 Da. All proteins with identified monolinks were included in a FASTA file for the main crosslinking search.

For the main searches settings similar to the initial search were applied with trypsin as the digestion enzyme, an MS1 error tolerance of ±15 ppm, and an MS2 error tolerance of ±0.2 Da. Crosslinked amino acids for DSS crosslinking were lysines and the protein N-terminus, for PDH aspartate and glutamate and for DMTMM lysines with aspartate and lysine with glutamate. The error tolerances were changed for the measurements that employed orbitrap detection. MS2 tolerance, crosslink MS2 tolerance, and peak matching tolerances (cp_tolerance, cp_tolerancexl) were set to ±15 ppm. The minimum peak number was set to 10.

Crosslinked peptides were filtered with a mass tolerance window of ±5 ppm, TIC ≥ 0.1, minimally 3 matched ions and a delta score of <0.9. To achieve <5% false discovery rate, the xQuest score cutoff was set at less than 5% of decoy hits compared to target hits. All spectra were checked and filtered manually with at least three bond cleavages. For AC8 heteromeric/inter-protein XLs, homomeric/intra-protein XLs, monolinks and homomultimeric links/intermolecular self-links (dimerisation liks) were exported. For the interactors we only exported hetero- and homomeric XLs, as well as homomultimeric links (dimerisation links). Crosslinks between 2 peptides sharing the same sequence were considered to be intermolecular selflinks.

### Data visualisation

All DSS, PDH and DMTMM crosslinks were combined in one file and used for visualisation in pyMOL v. 2.4.1 using the plugin PyXlinkViewer (Schiffrin et al, 2020). 30 Å were set as violation cutoff. This cutoff was defined based on the maximal Cα-Cα distance reported for DSS (Merkley et al, 2014). Distances calculated by PyXlinkViewer were exported and used for visualisation with xiNET (Combe et al, 2015).

## AC8-Gβγ docking

For docking studies, we used the Gβγ structure determined by Davis et al (Davis et al, 2005) (PDB: 1XHM). All three crosslinks between structured regions of AC8 and Gβγ were considered for docking.

HADDOCK 2.4 (van Zundert and Bonvin, 2014) was used to predict the AC8-Gβγ complex using the detected inter-protein crosslinks as distance restraints, keeping the default settings for protein-protein docking. DisVis (van Zundert and Bonvin, 2015) was used to visualize the accessible interaction space, check for false positives and to further calculate the residues involved in the interaction. The distances set in the restraints file were 8.7–20.5 Å for PDH and 6.5–16 Å for DMTMM (Leitner et al, 2014a). The residues used for docking were residue 674 and 678 on AC8 (PDH) and AC8 residue 936 (DMTMM). All 3 crosslinked with residue 312 on the Gβ subunit. GetArea (Fraczkiewicz and Braun, 1998) was used to calculate the per residue surface accessibility of AC8 and Gβγ and residues with at least 40% accessibility were considered for the analysis.

Cluster 1 showed the lowest mean van der Waals energy, the highest fraction of common contacts and a plausible orientation of Gβγ that would facilitate membrane anchoring (Appendix Fig. S11). That is why cluster 1, structure 1 was used for illustration purposes.

## AlphaFold-Multimer prediction

An AlphaFold-Multimer (AlphaFold v. 2.2.0) prediction of AC8 (UniProt ID: E1BQ12) interacting with CaM (UniProt ID: P62157) and Gαs (UniProt ID: P04896) was run on the Euler scientific computing cluster (ETH Zurich) using the standard submission script for SLURM submission and resource calculation (https://gitlab.ethz.ch/sis/alphafold_on_euler).

## Data availability

The coordinates have been deposited in the Protein Data Bank, with entry codes 8BUZ, 8BV5. The cryo-EM density maps have been deposited in the Electron Microscopy Data Bank, with accession numbers EMD-16249, EMD-16252, EMD-16253, EMD-16254 and EMD-16255. The mass spectrometry proteomics data have been deposited to the ProteomeXchange Consortium via the PRIDE partner repository with the dataset identifiers PXD040303 (LiP-MS), PXD040374 (XL-MS) and PXD044766 (PRM). The AC8-Gβγ docking model has been deposited to PDB-DEV (PDBDEV_00000214) and the AlphaFold2 model has been deposited to ModelArchive (10.5452/ma-gqm71). The R code used for data analysis is on GitHub (https://github.com/dschust-r/AC8_LiP_MS).

## Peer review information

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

## Acknowledgements

We thank Emiliya Poghosyan (EM Facility, PSI) and Miroslav Peterek (ScopeM, ETH Zurich) for expert support in cryo-EM data collection. We also thank Spencer Bliven and Marc Caubet-Serrabou (PSI) for the support in high performance computing. We thank Natalie de Souza and Jacopo Marino for valuable feedback on the manuscript. We thank Jonas Mühle for assistance with flashBAC expression system. We thank the Wollscheid lab at ETH Zurich for allowing us to acquire data on their mass spectrometer. The work was supported by the Swiss National Science Foundation grant 184951 (VMK), and by a grant from the Vontobel foundation (VMK). P.P. is supported by the EPIC-XS Consortium (grant agreement no. 823839).

## Author contributions

**Basavraj Khanppnavar**: Conceptualization; Data curation; Formal analysis; Validation; Investigation; Visualization; Methodology; Writing—original draft; Writing—review and editing. **Dina Schuster**: Conceptualization; Data curation; Formal analysis; Validation; Investigation; Visualization; Methodology; Writing —original draft; Writing—review and editing. **Pia Lavriha**: Data curation; Formal analysis; Investigation; Methodology; Writing—review and editing. **Federico Uliana**: Data curation; Formal analysis; Validation; Investigation; Methodology; Writing—review and editing. **Merve Özel**: Data curation; Formal analysis; Investigation; Methodology. **Ved Mehta**: Data curation; Formal analysis; Investigation; Methodology. **Alexander Leitner**: Resources; Data curation; Formal analysis; Supervision; Validation; Investigation; Methodology; Writing—review and editing. **Paola Picotti**: Resources; Supervision; Funding acquisition. **Volodymyr M Korkhov**: Conceptualization; Resources; Data curation; Formal analysis; Supervision; Funding acquisition; Validation; Investigation; Visualization; Methodology; Writing—original draft; Project administration; Writing—review and editing.

## Disclosure and competing interests statement

The authors declare no competing interests.

# Expanded View Figures

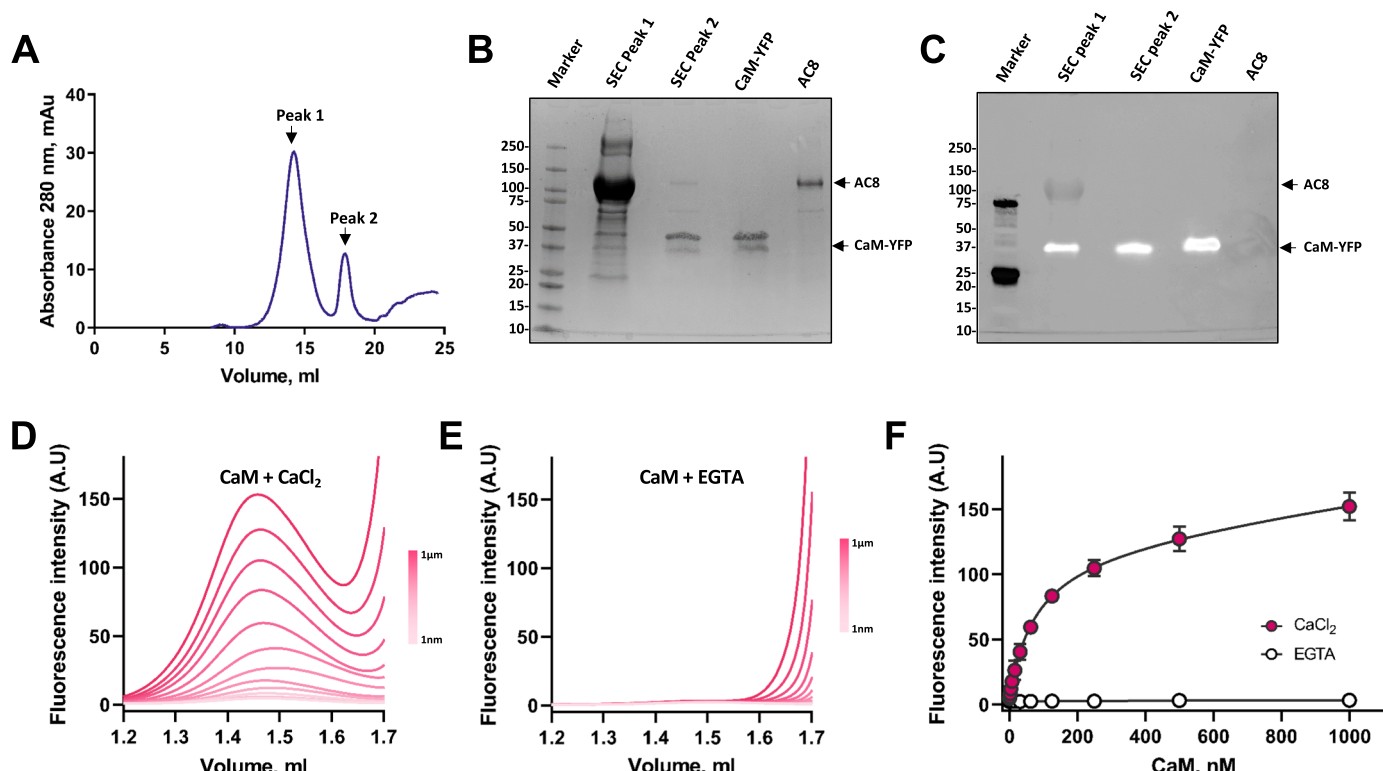

**Figure EV1. Purified AC8 binds CaM with high affinity in the presence of $Ca^{2+}$.**

(A) The size exclusion chromatography (SEC) profile of the AC8-$Ca^{2+}$/CaM-YFP complex. (B, C) SDS-PAGE analysis of AC8-$Ca^{2+}$/CaM-YFP complex. The complex was prepared by mixing purified recombinant AC8 and CaM-YFP in 1:2 ratio, respectively. The panel (B) shows the Coomassie stained SDS-PAGE of AC8-$Ca^{2+}$/CaM complex after SEC. Panel (C) shows overlayed prestained and YFP-fluorescence images of the SDS-PAGE displayed in panel (B). (D, E) Representative FSEC profiles an increase in fluorescence intensity of the AC8 elution peak with increasing concentration of CaM. (F) The saturation-binding curves show nanomolar affinity ($77.7 \pm 10.9$ nM, $n = 4$, technical replicates) of AC8 for CaM in presence of calcium and no AC8-CaM interaction in absence of calcium. For the experiments in panel (F), the data are shown as mean $\pm$ S.E.M. ($n = 4$). Source data are available online for this figure.

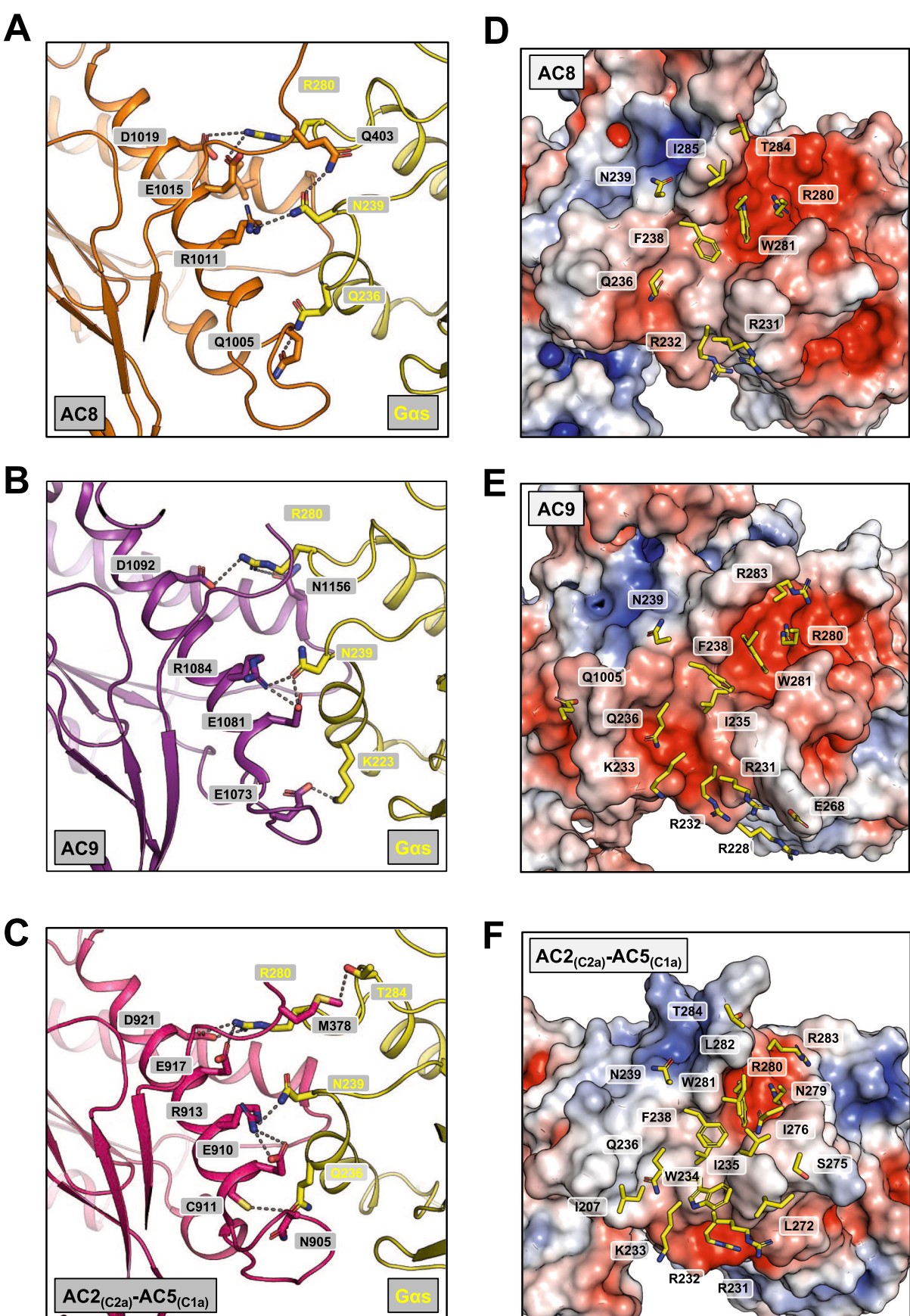

**Figure EV2.  Gαs binding interfaces in ACs.**

(A–C) Depiction of AC-Gαs interfacial residues observed in the cryo-EM structure of AC8-CaM-Gαs (top), AC9-Gαs (middle), and crystal structure of chimeric AC2($_{C2a}$)-AC5($_{C1a}$)-Gαs complex (bottom). (D–F) Electrostatic surface representation of AC8 (top), AC9 (middle), and chimeric AC2($_{C2a}$)-AC5($_{C1a}$) (bottom) showing interfacial residues of Gαs (yellow colour) within 4 Å of AC.

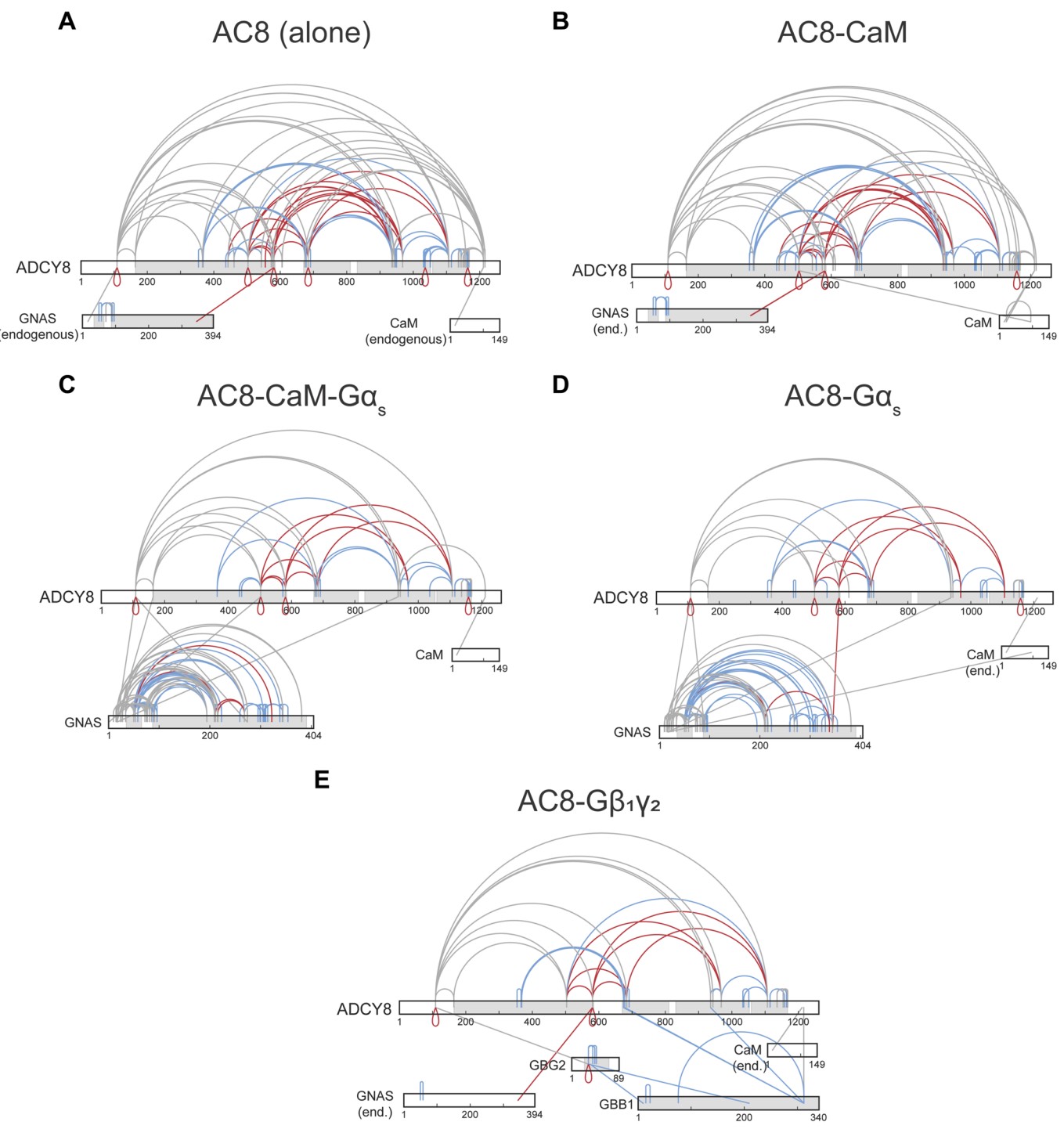

**Figure EV3. All detected crosslinks (DSS and PDH/DMTMM).**

(**A**) Crosslinked AC8 with copurified Gαs and CaM. Violated crosslinks (>30 Å) are coloured in red, satisfied crosslinks between structured/resolved regions are coloured in blue. Crosslinks that involve flexible regions and regions that are not resolved in the cryo-EM or in the structures used for protein-protein docking are coloured in grey. Homomultimeric links (oligomerisation links) are indicated with red drops. Regions with available structure are highlighted in grey on the respective protein sequence. (**B**) Crosslinked AC8 and CaM with copurified Gαs. (**C**) Crosslinked AC8, CaM and Gαs. (**D**) crosslinked AC8 and Gαs with copurified CaM (**E**) crosslinked AC8 and Gβγ with copurified Gαs and CaM.

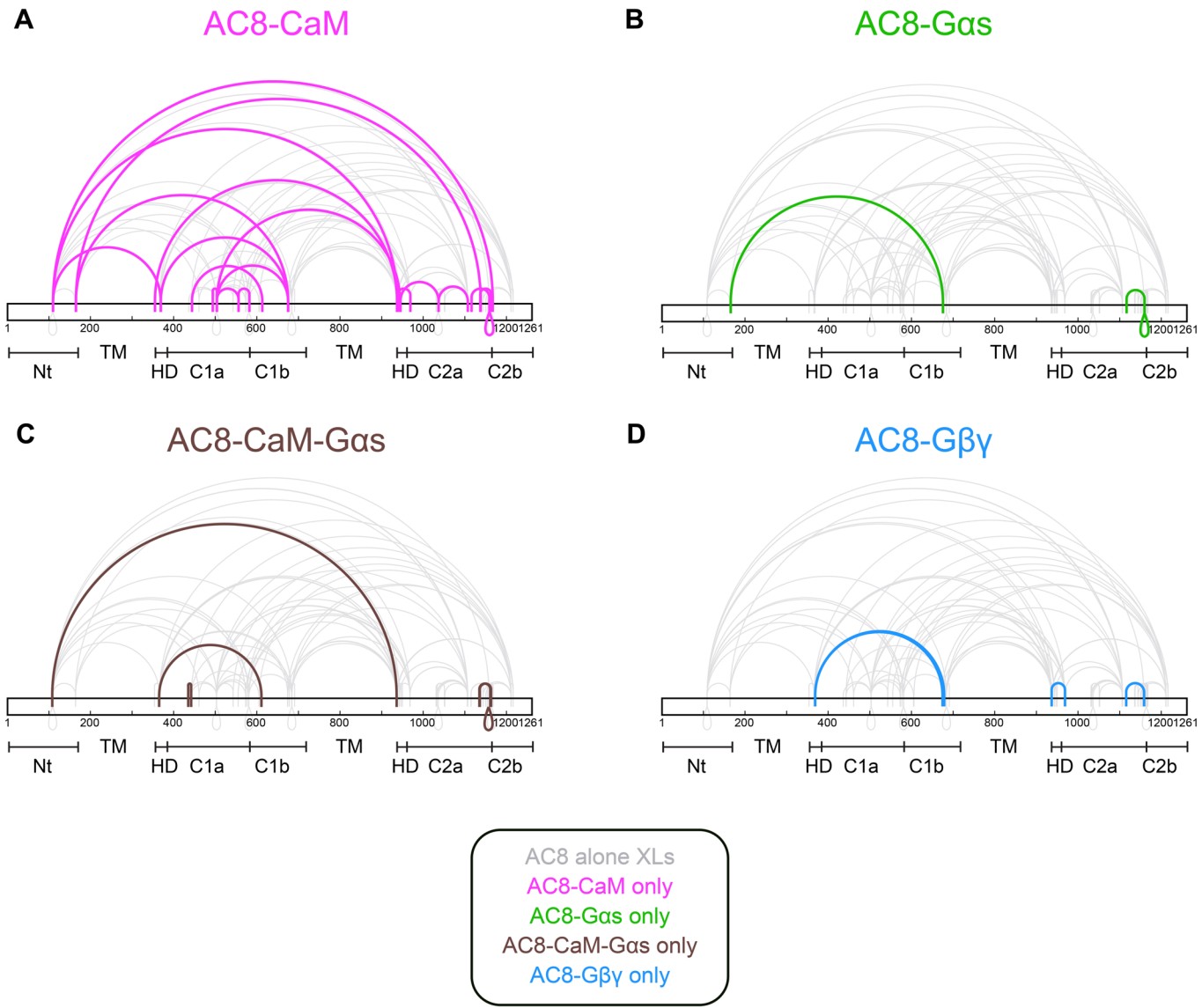

**Figure EV4. XL-MS comparison of identified crosslinks between AC8 by itself and interactors.**

(A) Crosslinks detected in the AC8 only sample (no added interactor) are coloured in light grey, positioned in the background. Crosslinks only found in the AC8-CaM sample (not in the AC8 alone sample) are coloured in pink. AC8 domains and their respective locations are indicated below. (B) Crosslinks only found in the AC8-Gαs sample are coloured in green. (C) Crosslinks identified only in the AC8-CaM-Gαs sample are highlighted in brown. (D) Crosslinks unique to the AC8-Gβγ sample are coloured in blue.

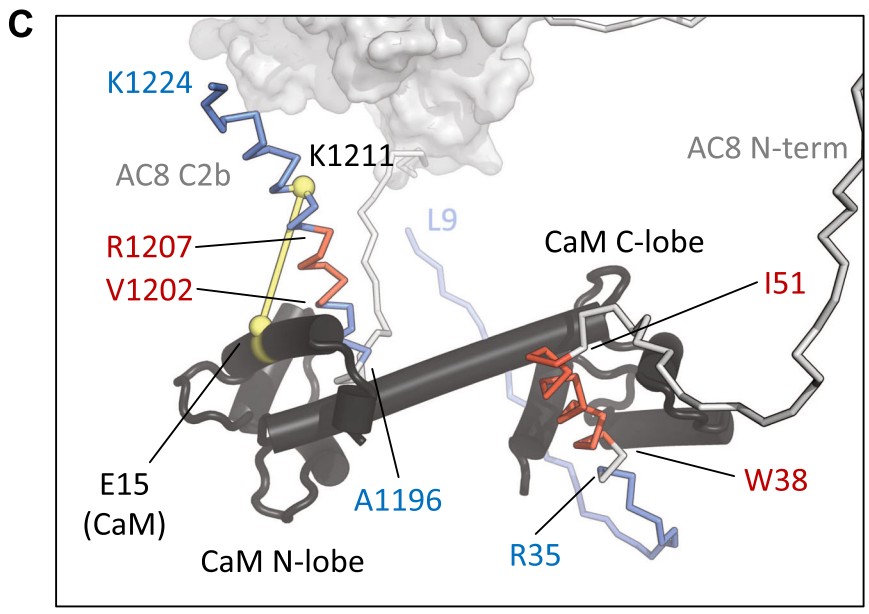

◄ **Figure EV5. AlphaFold2 model of AC8-CaM-Gαs.**

(A) The views of the model of the complex generated using AlphaFold2 as described in Materials and Methods, coloured according to the pLDDT scores (blue – high pLDDT / confidence; red – low pLDDT / confidence). (B) The same views as in a, with key elements of the predicted structure labelled, and individual proteins coloured white (AC8), pink (CaM) and green (Gαs). (C) The AlphaFold2 prediction matches well the known CaM binding sites (the 1-5-8-14 motif (WXXXVXXIXXXXXI) residues 38–51; the IQ-like motif (VQXXXR) 1202-1207, red (MacDougall et al, 2009)) and the LiP-MS-detected CaM binding peptides (9-35, 1196-1224, blue), respectively, in the N-terminus and C2b domain of AC8. CaM is coloured dark grey. The crosslink between E15 (CaM) and K1211 (AC8) detected in our XL-MS analysis is indicated as a yellow line. The distance between the Cα atoms of these residues (yellow spheres) is 17.5 Å.

