## [Peer Review File · EMBO Reports]

Regulatory sites of CaM-sensitive adenylyl cyclase AC8 revealed by cryo-EM and structural proteomics

Volodymyr Korkhov, Basavraj Khanppnavar, Dina Schuster, Pia Lavriha, Federico Uliana, Merve Özel, Ved Mehta, Alexander Leitner, and Paola Picotti

Corresponding author(s): Volodymyr Korkhov (volodymyr.korkhov@psi.ch)

Review Timeline:

Transfer Date:	18th Oct 23
Editorial Decision:	19th Oct 23
Revision Received:	3rd Dec 23
Editorial Decision:	8th Dec 23
Revision Received:	17th Jan 24
Accepted:	19th Jan 24

Editor: Achim Breiling

Transaction Report: This manuscript was transferred to EMBO reports following peer review at The EMBO Journal.

Dear Prof. Korkhov,

Thank you for transferring your revised manuscript to EMBO reports. I now went through the manuscript and the referee report from The EMBO Journal (attached again below). The referee has remaining concerns and suggestions to improve the manuscript, or to strengthen the data and the conclusions drawn.

Given the constructive referee comments, I would like to invite you to revise your manuscript with the understanding that the remaining concerns of the referee must be addressed in the revised manuscript or in a final point-by-point response.

In this case, please address the minor concerns of the referee. Moreover, please tone down or discuss further conclusions claiming that the reported structure is indeed calmodulin-bound AC8, as indicated by the referee in his/her report (part 'major concern').

Moreover, the manuscript needs formatting according to our journal style. Please carefully review the instructions that follow below.

When submitting your final revised manuscript, we will require:

- 1) a .docx formatted version of the final manuscript text (including legends for main figures, EV figures and tables), but without the figures included. Figure legends should be compiled at the end of the manuscript text.
- 2) individual production quality figure files as .eps, .tif, .jpg (one file per figure), of main figures and EV figures. Please upload these as separate, individual files upon re-submission.

The Expanded View format, which will be displayed in the main HTML of the paper in a collapsible format, has replaced the Supplementary information. You can submit up to 5 images as Expanded View. Please follow the nomenclature Figure EV1, Figure EV2 etc. The figure legend for these should be included in the main manuscript document file in a section called Expanded View Figure Legends after the main Figure Legends section. Additional Supplementary material should be supplied as a single pdf file labeled Appendix. The Appendix should have page numbers and needs to include a table of content on the first page (with page numbers) and legends for all content. Please follow the nomenclature Appendix Figure Sx, Appendix Table Sx etc. throughout the text, and also label the figures and tables according to this nomenclature. In this case, I think it will be possible to combine some of the Supplementary Figures to have in the end 5 EV figures. It is not necessary to have one EV figure related to one main figure. Just make sure that the panels are called out correctly.

- 3) a .docx formatted letter INCLUDING the reviewers' report and your detailed point-by-point responses to the remaining comments. As part of the EMBO Press transparent editorial process, the point-by-point response is part of the Review Process File (RPF), which will be published alongside your paper.
- 4) a complete author checklist, which you can download from our author guidelines (<https://www.embopress.org/page/journal/14693178/authorguide>). Please insert page numbers in the checklist to indicate where the requested information can be found in the manuscript. The completed author checklist will also be part of the RPF.

- 5) that primary datasets produced in this study (e.g. RNA-seq, ChIP-seq, structural and array data) are deposited in an appropriate public database. If no primary datasets have been deposited, please also state this in a dedicated section (e.g. 'No primary datasets have been generated and deposited'), see below.

The accession numbers and database should be listed in a formal "Data Availability" section (placed after Materials & Methods) that follows the model below. This is now mandatory (like the COI statement). Please note that the Data Availability Section is restricted to new primary data that are part of this study. This section is mandatory. As indicated above, if no primary datasets have been deposited, please state this in this section

Data availability

8) Regarding data quantification and statistics, please make sure that the number "n" for how many independent experiments were performed, their nature (biological versus technical replicates), the bars and error bars (e.g. SEM, SD) and the test used to calculate p-values is indicated in the respective figure legends (also for potential EV and Appendix figures). Please also check that all the p-values are explained in the legend, and that these fit to those shown in the figure. Please provide statistical testing where applicable. Please avoid the phrase 'independent experiment', but clearly state if these were biological or technical replicates. Please also indicate (e.g. with n.s.) if testing was performed, but the differences are not significant. In case n=2, please show the data as separate datapoints without error bars and statistics. See also: <http://www.embopress.org/page/journal/14693178/authorguide#statisticalanalysis>

9) Please also note our reference format:

10) We updated our journal's competing interests policy in January 2022 and request authors to consider both actual and perceived competing interests. Please review the policy <https://www.embopress.org/competing-interests> and update your competing interests if necessary. Please name this section 'Disclosure and Competing Interests Statement' and put it after the Acknowledgements section.

11) We now use CRediT to specify the contributions of each author in the journal submission system. CRediT replaces the author contribution section. Please use the free text box to provide more detailed descriptions and do NOT add an author contributions section to the manuscript text file. See also guide to authors: <https://www.embopress.org/page/journal/14693178/authorguide#authorshippinguidelines>

12) Please add up to 5 keywords to the manuscript and order the manuscript sections like this, using these names: Title page - Abstract - Keywords - Introduction - Results - Discussion - Materials and Methods - Data availability section - Acknowledgements - Disclosure and Competing Interests Statement - References - Figure legends - Expanded View Figure legends - Tables

13) Please enter all the funding information also into our submission system and make sure this is complete and similar to the one mentioned in the manuscript text file.

14) Please provide all material and methods information in the main manuscript text file. Moreover, we would encourage you to use 'Structured Methods', our new Materials and Methods format. According to this format, the Materials and Methods section should include a Reagents and Tools Table (listing key reagents, experimental models, software and relevant equipment and including their sources and relevant identifiers) followed by a Methods and Protocols section in which we encourage the authors to describe their methods using a step-by-step protocol format with bullet points, to facilitate the adoption of the methodologies across labs. More information on how to adhere to this format as well as downloadable templates (.doc or .xls) for the Reagents and Tools Table can be found in our author guidelines (section 'Structured Methods'):

In addition, I would need from you:

I look forward to seeing a revised version of your manuscript when it is ready. Please let me know if you have questions or comments regarding the revision.

Kind regards,

Achim

Referee #1:

Mammalian adenylyl cyclases play the key roles in integrating diverse extracellular and intracellular signals to generate diffusible second messenger, cAMP for the diverse physiological functions. Despite the cloning and characterizations for the regulations of nine isoforms of adenylyl cyclases in late 80 and early 90, the structural basis for the regulation of this family of enzymes was limited to the engineered soluble adenylyl cyclase, which is consist of conserved C1a and C2a domains of this enzyme until the report of cryoEM structure of AC9 done by the same group of this manuscript. This report provides the cryoEM structure of the second mammalian adenylyl cyclase, AC8, which has the regulation that is distinct from AC9, particularly for the regulation by calcium-calmodulin and G protein betagamma subunit. Authors combine cryoEM and biochemical studies to show the overall structure of AC8 and the regulation of this isoform by four known regulators, G α , G $\beta\gamma$, calmodulin and forskolin. Overall, the work is technically sound and data shown here provide new insights how AC8 is regulated, which may be applicable to other adenylyl cyclases. The revised manuscript has provided some updates that have improved the manuscript. However, the major concern on whether the reported structure is indeed calmodulin-bound AC8 remains unresolved. While authors make some effort to discuss this issue in the discussion, readers will be misled from reading the abstract that it is indeed calmodulin-bound AC8 structure even though no calmodulin can be seen and no structural insight how calmodulin activates AC8 is revealed by the structure. My specific comments are below that should also be addressed.

Major concern:

Is CaM presence in the AC8-CaM-G α complex for cryoEM analysis? CaM cannot be modeled into the Coulomb potential map thus is absent in the final models of cryoEM single particle reconstruction of the presumed AC8-G α -CaM complex. Authors provide the possible explanation that its absence is due to the flexibility where CaM binds, the N terminus and C2b region of AC8. Authors also provide the additional experimental data to show the interaction between YFP-CaM and AC8. However, none of such data can exclude the possibility that the pre-mixed AC8-G α -CaM-forskolin, which did not go through SEC does not form the stable enough complex to survive the grid making process so that CaM is NOT present in the final particle on the grids. This is not consistent with the statement in the abstract "Here we report the 3.5 Å resolution cryo-EM structure of the bovine AC8 bound to Ca²⁺/CaM and the stimulatory G α s protein". Most of readers will read this statement in abstract to believe that the reported structure is indeed a CaM- and G α -bound AC8 structure.

Minor concerns:

1. In the previous version, I comment that " Cryo EM data for AC8 in detergent has the profound orientation bias (figure s2d) evidenced by the holes in the angular distribution. Such distribution is not shown for AC8 in lipid nanodisc. Authors should report sphericity based on Tan et al Nature Methods 2018 or new measurement to support the missing angles does not profoundly bias the map. Authors provide a new angular distribution map where the holes are now gone. But, it is unclear from this reviewer how the holes are gone without the new data collection. Since only an 180-degree view is provided in supplemental figure 7 and giving the fact that there were visible holes in the previous version of manuscript, this reviewer wonders whether such holes are now on the back so that it is not visible from the front view. In addition, they claim in their response that they have included the sphericity data in revised manuscript by citing supplemental figure 7. However, the sphericity data is not reported in the supplemental figure 7.

2. In the discussion, authors state that "However, specifically in the AC field interesting parallels can be drawn in the regulatory mechanisms of AC8 and the bacterial adenylyl cyclase toxins, Bordetella pertussis CyaA (44) and the Bacillus anthracis oedema factor (45). CaM has been found to interact with the disordered regions of these toxin ACs, allosterically communicating with the catalytic site and influencing the ordering of the catalytic site to stimulate the enzymatic activity." To my knowledge, the statement is not true. In case of anthrax edema factor (EF), the structures of EF alone and CaM-bound EF are available. Multiple regions of EF, including the helical domain, switch A, and switch C, and CA domain come together to embrace calcium-CaM. Since all four regions are resolved as the ordered structures in EF alone structure, one should not be able to claim that CaM binds the disordered regions of EF. In term of Bordetella pertussis CyaA, only c-CaM bound AC domain of cyaA is available. From that, the c-CaM interaction with CyaA is mainly mediated by several helices (helix F, H, H') and c-tail of cyaA AC domain, which is also mostly helix. Thus, the CaM contact region of cyaA should not be claimed to be disordered region as well.

3. Authors generate a model in figure 7 to show that the extended conformation of CaM binds AC8, which is not the conventional CaM binding to their cellular targets where N-CaM and C-CaM come together to bind its cellular targets. They should provide the evidence for such modeling if they would like to be so specific.

EMBOR-2023-58339-T

Response to reviewers' comments

Referee #1:

Mammalian adenylyl cyclases play the key roles in integrating diverse extracellular and intracellular signals to generate diffusible second messenger, cAMP for the diverse physiological functions. Despite the cloning and characterizations for the regulations of nine isoforms of adenylyl cyclases in late 80 and early 90, the structural basis for the regulation of this family of enzymes was limited to the engineered soluble adenylyl cyclase, which consists of conserved C1a and C2a domains of this enzyme until the report of cryoEM structure of AC9 done by the same group of this manuscript. This report provides the cryoEM structure of the second mammalian adenylyl cyclase, AC8, which has the regulation that is distinct from AC9, particularly for the regulation by calcium-calmodulin and G protein betagamma subunit. Authors combine cryoEM and biochemical studies to show the overall structure of AC8 and the regulation of this isoform by four known regulators, G α , G $\beta\gamma$, calmodulin and forskolin. Overall, the work is technically sound and data shown here provide new insights how AC8 is regulated, which may be applicable to other adenylyl cyclases. The revised manuscript has provided some updates that have improved the manuscript. However, the major concern on whether the reported structure is indeed calmodulin-bound AC8 remains unresolved. While authors make some effort to discuss this issue in the discussion, readers will be misled from reading the abstract that it is indeed calmodulin-bound AC8 structure even though no calmodulin can be seen and no structural insight how calmodulin activates AC8 is revealed by the structure. My specific comments are below that should also be addressed.

Major concern:

Is CaM presence in the AC8-CaM-G α complex for cryoEM analysis? CaM cannot be modeled into the Coulomb potential map thus is absent in the final models of cryoEM single particle reconstruction of the presumed AC8-G α -CaM complex. Authors provide the possible explanation that its absence is due to the flexibility where CaM binds, the N terminus and C2b region of AC8. Authors also provide the additional experimental data to show the interaction between YFP-CaM and AC8. However, none of such data can exclude the possibility that the pre-mixed AC8-G α -CaM-forskolin, which did not go through SEC does not form the stable enough complex to survive the grid making process so that CaM is NOT present in the final particle on the grids. This is not consistent with the statement in the abstract "Here we report the 3.5 Å resolution cryo-EM structure of the bovine AC8 bound to Ca²⁺/CaM and the stimulatory Gas protein". Most of readers will read this statement in abstract to believe that the reported structure is indeed a CaM- and G α -bound AC8 structure.

RESPONSE: We understand this concern of the reviewer, and we agree that this needs to be clarified. To ensure that the readers are given the correct information and are not led to any misinterpretation of our results, we have updated the abstract and the results section as follows.

Abstract: "Here we report the 3.5 Å resolution cryo-EM structure of the bovine AC8 bound to the stimulatory Gas protein in the presence of Ca²⁺/CaM".

Results, line 142: "Among the tested combinations, one sample proved to be suitable for structure determination: AC8 in the presence of Ca²⁺/CaM and activated Gas, MANT-GTP and forskolin. After multiple rounds of 2D and 3D classifications, the best set of particles resulted in a 3D reconstruction at a resolution of 3.5 Å (Fig EV4-6, Table 2).

The structure of AC8-CaM-Gas (AC8 bound to Gas in the presence of Ca²⁺/CaM) revealed..."

Minor concerns:

1. In the previous version, I comment that "Cryo EM data for AC8 in detergent has the profound orientation bias (figure s2d) evidenced by the holes in the angular distribution. Such distribution is not shown for AC8 in lipid nanodisc. Authors should report sphericity based on Tan et al Nature Methods 2018 or new measurement to support the missing angles does not profoundly bias the map. Authors

provide a new angular distribution map where the holes are now gone. But, it is unclear from this reviewer how the holes are gone without the new data collection. Since only an 180-degree view is provided in supplemental figure 7 and giving the fact that there were visible holes in the previous version of manuscript, this reviewer wonders whether such holes are now on the back so that it is not visible from the front view. In addition, they claim in their response that they have included the sphericity data in revised manuscript by citing supplemental figure 7. However, the sphericity data is not reported in the supplemental figure 7.

RESPONSE: The sphericity analysis has been included in the new EV Figures describing the cryo-EM workflows (Fig EV4 and Fig EV7). We have also added illustrations indicating angular distributions from different angles.

2. In the discussion, authors state that "However, specifically in the AC field interesting parallels can be drawn in the regulatory mechanisms of AC8 and the bacterial adenylyl cyclase toxins, *Bordetella pertussis* CyaA (44) and the *Bacillus anthracis* oedema factor (45). CaM has been found to interact with the disordered regions of these toxin ACs, allosterically communicating with the catalytic site and influencing the ordering of the catalytic site to stimulate the enzymatic activity." To my knowledge, the statement is not true. In case of anthrax edema factor (EF), the structures of EF alone and CaM-bound EF are available. Multiple regions of EF, including the helical domain, switch A, and switch C, and CA domain come together to embrace calcium-CaM. Since all four regions are resolved as the ordered structures in EF alone structure, one should not be able to claim that CaM binds the disordered regions of EF. In term of *Bordetella pertussis* CyaA, only c-CaM bound AC domain of cyaA is available. From that, the c-CaM interaction with CyaA is mainly mediated by several helices (helix F, H, H') and c-tail of cyaA AC domain, which is also mostly helix. Thus, the CaM contact region of cyaA should not be claimed to be disordered region as well.

RESPONSE: We thank the reviewer for highlighting this. To avoid confusion, we have revised the second sentence as follows (line 387):

"CaM binding has been found to allosterically influence the dynamics of the catalytic sites in these enzymes, stimulating the enzymatic activity."

3. Authors generate a model in figure 7 to show that the extended conformation of CaM binds AC8, which is not the conventional CaM binding to their cellular targets where N-CaM and C-CaM come together to bind its cellular targets. They should provide the evidence for such modeling if they would like to be so specific.

RESPONSE: The model in last figure (now Figure 5) is meant to illustrate the different interactions, based on cryo-EM, crosslinking, docking and AlphaFold2 predictions. The CaM model is based on the output of the AlphaFold2 prediction, and it is supported by our XL-MS and LiP-MS data. This figure is not meant to represent the experimentally observed EM density. Although there is some recent structural evidence that two lobes of CaM (N-lobe and C-lobes) can simultaneously bind two different calmodulin binding motifs (Singh et al, 2018, PMID: 30116787), we are using the AlphaFold2 prediction model as an illustration in this case – and we certainly would not overstate the meaning of this illustration. To avoid any confusion, we have clarified in the legend that this is model is shown for illustration purposes (line 1216):

"D-E An illustrative model of AC8 and its activating (G α s and CaM) or inhibitory (G $\beta\gamma$) interactors, based on cryo-EM (AC8-G α s-CaM), docking (AC8-G $\beta\gamma$) and AlphaFold2 (AC8-G α s-CaM)."

Dear Prof. Korkhov,

Thank you for the submission of your further revised manuscript to our editorial offices. I now went through the manuscript and your p-b-p-response, and I consider the remaining points by the referee as adequately addressed.

Before I can proceed with formal acceptance, I have these editorial requests I ask you to address in a final revised manuscript:

- Please upload a complete author checklist with your final submission, which you can download from our author guidelines: <https://www.embopress.org/page/journal/14693178/authorguide>

Please insert page numbers in the checklist to indicate where the requested information can be found in the manuscript. The completed author checklist will also be part of the RPF.

- Please make sure that all figure panels are called out separately and sequentially (main, EV and Appendix figures). Presently, it seems callouts for panels/Figs. 5D, 5E, 6D, 6G, 7E and EV15 are missing. Please check.

- We can publish only 5 EV figures online. Please select 5 EV figures for this and follow the nomenclature Figure EVx for these and their callouts. The additional supplementary material needs to be supplied in an Appendix. The Appendix needs to be a single pdf file labeled Appendix. The Appendix should have page numbers and needs to include a table of content on the first page (with page numbers) and legends for all content. Please follow the nomenclature Appendix Figure Sx, Appendix Table Sx etc. throughout the text, and also label the figures and tables according to this nomenclature. Please put the legends of the Appendix items below these in the Appendix and remove these from the main manuscript text file. Please make sure that all callouts are updated accordingly.

- I would suggest to move the three tables also to the Appendix (as Appendix Tables S1-3). Please do that and make sure that all callouts are updated accordingly. Then remove the tables from the manuscript text file.

- Please make sure that the number "n" for how many independent experiments were performed, their nature (biological versus technical replicates), the bars and error bars (e.g. SEM, SD) and the test used to calculate p-values is indicated in the respective figure legends (for main, EV and Appendix figures) of the final revised manuscript. Please also check that all the p-values are explained in the legend, and that these fit to those shown in the figure. Please provide statistical testing where applicable. Please avoid the phrase 'independent experiment', but clearly state if these were biological or technical replicates. Please also indicate (e.g. with n.s.) if testing was performed, but the differences are not significant. In case n=2, please show the data as separate datapoints without error bars and statistics. See also: <http://www.embopress.org/page/journal/14693178/authorguide#statisticalanalysis>

If n<5, please show single datapoints for diagrams. Presently, several diagrams miss statistical testing (e.g. 1B-F and 2D/E). Moreover:

- Please define for the box plots in Fig. EV15 the minima, maxima, centre, bounds of box and whiskers, and percentile in their legends of figures.

- Please provide information related to n in the legends of figures EV 15a-d.

- Although 'n' is provided, please describe the nature of 'n' (biological or technical) in the legends of figures 1b-f; 2d-e; EV 3f.

- Please define the error bars in the legends of figures 1b-c and EV 15d.

- I would suggest adding to each legend (of the main, EV and Appendix figures) a 'Data Information' section explaining the statistics used or providing information regarding replicates and scales. See:

- Please make sure that all the funding information is also entered into the online submission system and that it is complete and similar to the one in the acknowledgement section of the manuscript text file. Presently, a grant (?) by the Vontobel foundation (VMK) is only mentioned in the acknowledgements.

- It seems all the requested source data was provided. But could you please also provide a completed source data checklist (attached again).

Best,

EMBOR-2023-58339V3

Point-by-point response

Dear Dr. Breiling,

Please find below a point-by-point response to your comments, which we have used to reformat the manuscript, double check the source data consistency, add statistics where appropriate, which overall further improved the manuscript.

In preparation we made several changes as detailed below. It took us substantially longer than expected due to absences of the key authors, and due to an extensive checks and corrections. Please find below the details of the changes (with your points in blue):

- Please upload a complete author checklist with your final submission, which you can download from our author guidelines:<https://www.embopress.org/page/journal/14693178/authorguide> Please insert page numbers in the checklist to indicate where the requested information can be found in the manuscript. The completed author checklist will also be part of the RPF.

The updated checklist has been prepared. The location of the information in the manuscript has been included (sections, pages and lines).

- Please make sure that all figure panels are called out separately and sequentially (main, EV and Appendix figures). Presently, it seems callouts for panels/Figs. 5D, 5E, 6D, 6G, 7E and EV15 are missing. Please check.

We checked, and to the best of our knowledge this is now the case and the figures are called out consistently (they have new names, as some of the figures moved to the Appendix).

- We can publish only 5 EV figures online. Please select 5 EV figures for this and follow the nomenclature Figure EVx for these and their callouts. The additional supplementary material needs to be supplied in an Appendix. The Appendix needs to be a single pdf file labeled Appendix. The Appendix should have page numbers and needs to include a table of content on the first page (with page numbers) and legends for all content. Please follow the nomenclature Appendix Figure Sx, Appendix Table Sx etc. throughout the text, and also label the figures and tables according to this nomenclature. Please put the legends of the Appendix items below these in the Appendix and remove these from the main manuscript text file. Please make sure that all callouts are updated accordingly.

Following your suggestion, we have limited the EV figures to only 5. The rest of the figures are now in the Appendix. The corresponding Source Data also now refer to the Appendix figures.

The new figure structure is as follows, listing the previous figure and the corresponding revised ones side by side:

EV1 – Appendix Figure S1
EV2 – Appendix Figure S2
EV3 – EV1
EV4 – Appendix Figure S3
EV5 – Appendix Figure S4
EV6 – Appendix Figure S5
EV7 – Appendix Figure S6
EV8 – Appendix Figure S7
EV9 – Appendix Figure S8
EV10 – EV2
EV11 – Appendix Figure S9
EV12 – Appendix Figure S10
EV13 – EV3
EV14 – EV4
EV15 – Appendix Figure S11
EV16 – EV5

- I would suggest to move the three tables also to the Appendix (as Appendix Tables S1-3). Please do that and make sure that all callouts are updated accordingly. Then remove the tables from the manuscript text file.

The tables are now in the Appendix.

- Please make sure that the number "n" for how many independent experiments were performed, their nature (biological versus technical replicates), the bars and error bars (e.g. SEM, SD) and the test used to calculate p-values is indicated in the respective figure legends (for main, EV and Appendix figures) of the final revised manuscript. Please also check that all the p-values are explained in the legend, and that these fit to those shown in the figure. Please provide statistical testing where applicable. Please avoid the phrase 'independent experiment', but clearly state if these were biological or technical replicates. Please also indicate (e.g. with n.s.) if testing was performed, but the differences are not significant. In case n=2, please show the data as separate datapoints without error bars and statistics. See also: <http://www.embopress.org/page/journal/14693178/authorguide#statisticalanalysis>

If n<5, please show single datapoints for diagrams. Presently, several diagrams miss statistical testing (e.g. 1B-F and 2D/E).

We have updated the manuscript with this information, added the n number, performed the statistical analysis of the data in Figure 1 and 2.

Moreover:

- Please define for the box plots in Fig. EV15 the minima, maxima, centre, bounds of box and whiskers, and percentile in their legends of figures.

The box plot has been defined now in the legend of the corresponding figure (Appendix Fig. S11, page 16 of the appendix).

- Please provide information related to n in the legends of figures EV 15a-d.
The explanation of the datapoints and the clusters is now provided in the legend.

- Although 'n' is provided, please describe the nature of 'n' (biological or technical) in the legends of figures 1b-f; 2d-e; EV 3f.

We updated the text, these are technical replicates (repeats).

- Please define the error bars in the legends of figures 1b-c and EV 15d.
The error bars have been defined for all figures now.

- I would suggest adding to each legend (of the main, EV and Appendix figures) a 'Data Information' section explaining the statistics used or providing information regarding replicates and scales. See:

We have included the data information in each legend (as the text within the corresponding section of the legend).

- Please make sure that all the funding information is also entered into the online submission system and that it is complete and similar to the one in the acknowledgement section of the manuscript text file. Presently, a grant (?) by the Vontobel foundation (VMK) is only mentioned in the acknowledgements.

The grant from the Vontobel foundation will be duly acknowledged in the online system.

- It seems all the requested source data was provided. But could you please also provide a completed source data checklist (attached again).

I have completed the checklist, indicating which previous figures correspond to the new ones. All source data has been provided.

In the process of going through the source data and checking data consistency, we discovered that several data figures (some of the graphs in Figures 1 and 2). The careful examination of all source data revealed that some of the data used an incorrect baseline value, due to copy-paste error of one fixed value. We took pains to carefully check all of the data and to correct this - the revised figures, source data are consistent and have been verified. As mentioned above, statistical tests were performed with all data as suggested (one way ANOVA). The modifications in figures 1-2 concerned the following panels:

Figure 1D, 2D and 2E: We mistakenly used wrong experimental blanks (i.e. blank from experimental replicate 1) for analysis of replicate 2 & 3. This was due to a copy-paste error in the excel sheet that was used for calculations (provided in the source data file. All datasets have been checked, the identified errors have been corrected in the source data files and in the corresponding figure panels.

Figure 1B & 1C: Selected data points from these two graphs (the identical treatments / conditions performed in different experiments) were merged in a new

analysis to improve the overall statistics. The revised figure panels show not only the bar graphs, but also the individual data points as circles. One outlier was detected for AC8 vs FSK condition (replicate 4 in Fig 1B and replicate 8 in Fig 1C) based on ROUT test ($Q = 1\%$).

I hope that the revised version of the manuscript is acceptable for publication, and I will be happy to provide any additional information if necessary.

Best regards,
Volodymyr Korkhov

Prof. Volodymyr Korkhov
Paul Scherrer Institute
OSRA/001
Villigen, CH-AG 5232
Switzerland

Dear Prof. Korkhov,

I am very pleased to accept your manuscript for publication in the next available issue of EMBO reports. Thank you for your contribution to our journal.

Yours sincerely,
